# Associate toxin-antitoxin with CRISPR-Cas to kill multidrug-resistant pathogens

Rui Wang [1,2,8], Xian Shu [1,3,8], Huiwei Zhao [1,8], Qiong Xue[1], Chao Liu[1,3], Aici Wu [1,3], Feiyue Cheng [1], Lingyun Wang[1,4], Yihan Zhang[1,5], Jie Feng[3,6], Nannan Wu [7] & Ming Li [1,3] ✉

CreTA, CRISPR-regulated toxin-antitoxin (TA), safeguards CRISPR-Cas immune systems by inducing cell dormancy/death upon their inactivation. Here, we characterize a bacterial CreTA associating with the I-F CRISPR-Cas in *Acinetobacter*. CreT is a distinct bactericidal small RNA likely targeting several essential RNA molecules that are required to initiate protein synthesis. CreA guides the CRISPR effector to transcriptionally repress CreT. We further demonstrate a proof-of-concept antimicrobial strategy named ATTACK, which AssociaTes TA and CRISPR-Cas to Kill multidrug resistant (MDR) pathogens. In this design, CRISPR-Cas is programed to target antibiotic resistance gene(s) to selectively kill MDR pathogens or cure their resistance, and when CRISPR-Cas is inactivated or suppressed by unwanted genetic or non-genetic events/factors, CreTA triggers cell death as the last resort. Our data highlight the diversity of RNA toxins coevolving with CRISPR-Cas, and illuminate a combined strategy of CRISPR and TA antimicrobials to 'ATTACK' MDR pathogens.

Antibiotic resistance (AR) is posing a serious threat to the global health. By accumulating intrinsic resistant mutations or horizontally transferred AR genes, bacteria can evolve to be multidrug-resistant (MDR), extensively drug-resistant or even pandrug-resistant[1]. On the 2017 WHO global priority list of AR pathogens, carbapenem-resistant *Enterobacteriaceae*, carbapenem-resistant *Pseudomonas aeruginosa*, and carbapenem-resistant *Acinetobacter baumannii* (CRAB) are ranked in the highest priority category[2].

Because the development of new antibiotics is far slower than the emergence of AR pathogens, there is an urgent need to develop novel antimicrobial strategies. The antimicrobials based on CRISPR (Clustered Regularly Interspaced Short Palindromic Repeats) have been proposed to be a promising strategy to combat antibiotic resistance[3,4]. CRISPR is an array that stores short invading (sometimes genomic) DNA fragments as spacer sequences intervening its repeat units. Its RNA products, namely CRISPR RNAs (crRNAs), guide Cas (CRISPR-associated) proteins to recognize and destruct the re-infecting foreign DNA/RNA, thus providing adaptive immunity[5–8]. CRISPR-Cas systems are highly diversified and hitherto classified into two major classes, with the immune effector being a multi-subunit complex (Class 1) or a single nuclease (Class 2), and further divided into 6 types and more than 30 subtypes[9]. The CRISPR antimicrobial consists of a crRNA that specifies a target sequence on the bacterial genome (usually selected from AR genes) and a Cas effector that generates a double strand break (DSB) at the target site[3,4,10,11]. As a result, the vast majority of pathogens will die with only a small fraction surviving by mutating the AR gene. Therefore, CRISPR antimicrobials can specifically eradicate AR pathogens in a complex bacterial community or cure their antibiotic resistance, and have been designed for target elimination or re-sensitization of different MDR pathogens, like *Escherichia coli*[3,10,12–14], *Staphylococcus aureus*[4], *Clostridioides difficile*[11], and *Enterococcus faecalis*[15].

[1]CAS Key Laboratory of Microbial Physiological and Metabolic Engineering, State Key Laboratory of Microbial Resources, Institute of Microbiology, Chinese Academy of Sciences, Beijing, China. [2]Non-coding RNA and Drug Discovery Key Laboratory of Sichuan Province, Chengdu Medical College, Chengdu, Sichuan, China. [3]College of Life Science, University of Chinese Academy of Sciences, Beijing, China. [4]College of Plant Protection, Shandong Agricultural University, Taian, Shandong, China. [5]School of life Sciences, Hebei University, Baoding, Hebei, China. [6]State Key Laboratory of Microbial Resources, Institute of Microbiology, Chinese Academy of Sciences, Beijing, China. [7]Shanghai Institute of Phage, Shanghai Public Health Clinical Center, Fudan University, Shanghai, China. [8]These authors contributed equally: Rui Wang, Xian Shu, Huiwei Zhao. ✉e-mail: lim_im@im.ac.cn

However, bacterial resistance to CRISPR antimicrobials can evolve by mutating the target sequence or by inactivating the CRISPR-Cas machinery, with the latter far more likely to occur[3,4,16]. CRISPR antimicrobials can be inactivated by spontaneous mutations, chromosomal rearrangements, or mobile genetic elements that destroy the *cas* genes, or by deletion of the AR-targeting CRISPR spacer. Besides, the diverse and likely ubiquitous anti-CRISPR (*acr*) genes, which encode small proteins to inactivate critical Cas proteins[17,18], can also confer on pathogens resistance to CRISPR antimicrobials. Therefore, more stable CRISPR-Cas systems resistant to these unwanted genetic variations and Acr proteins are required to develop robust CRISPR antimicrobials.

Our lab recently uncovered that CRISPR-regulated toxin-antitoxin (CreTA) modules associate with and safeguard diverse Class 1 CRISPR-Cas systems (encoding a multi-subunit immune effector)[19]. In *Haloarcula hispanica* where the first CreTA module was characterized, CreT is a bacteriostatic RNA which arrests cellular growth by sequestering a rare tRNA species, while CreA is a degenerated variant of crRNA which reprograms the CRISPR immune effector to transcriptionally repress *creT*. Therefore, CreTA is a two-RNA toxin-antitoxin (TA) system, which was subsequently considered to represent a distinct TA type, type VIII[20]. Our previous data showed that CreTA can elicit cell dormancy/death when *cas* genes are disrupted by active insertion (IS) elements, thus making bacterial cells addicted to a functional CRISPR effector. We consider CreTA to be a 'broad-spectrum' anti-anti-CRISPR mechanism, which can defend CRISPR-Cas at the population level against any Acr proteins that inactivate the effector protein(s) and various genetic variations that destruct the encoding *cas* gene(s), and possibly can also ensure the substantial expression and activity of CRISPR-Cas. So, we conceive that such TA modules can be harnessed to improve the stability and performance of CRISPR antimicrobials, and in this study propose an antimicrobial strategy, namely ATTACK, to AssociaTe TA and CRISPR-Cas to Kill MDR pathogens.

The key of ATTACK strategy is to find a CreTA module suitable for the CRISPR-Cas effector programed to kill pathogenic bacteria. Unfortunately, CreTA modules prove to be highly specific to their physically-linked CRISPR-Cas loci, and the hitherto experimentally validated CreTA modules all derive from archaeal species and should be difficult to be adapted to fitting bacterial CRISPR-Cas effectors[21,22]. Hence, in this study, we first endeavored to identify and characterize a bacterial CreTA, which lurks in a subtype I-F CRISPR-Cas system in *Acinetobacter*. By characterizing this CreTA module, we uncovered a distinct bactericidal small RNA, which likely disrupts protein synthesis. According to our previous study on the determinants of CreTA specificity[22], we engineered this CreTA module to make it compatible with the CRISPR-Cas effector of *A. baumannii* AYE, and then integrated them into a robust and effective antimicrobial to eradicate/cure MDR pathogens.

## Results

### Search for *creTA* elements associating with I-F CRISPR-Cas

Previously identified *creTA* genes all surround the *cas6* gene of a type I or III CRISPR-Cas[19]. Yet, within the I-F CRISPR-Cas loci of *Acinetobacter* species, we did not find any *creTA*-like elements by searching the sequences surrounding *csy4* (a subtype-specific *cas6* gene). We noticed that the *cas* operon is continuous in most *Acinetobacter* species (like the case of *A. baumannii* AYE depicted in Fig. 1a), but in *Acinetobacter* sp. WCHA45, LoGeW2-3, and ANC3789, *cas3* and *csy1* are separated by a ~400 bp intergenic region (IGR) (Fig. 1a). Interestingly, these three IGRs share marked sequence similarities (Fig. 1b). We also noted that CRISPR arrays from these three species share nearly identical repeat sequences (of which 40% nucleotides differ from the AYE CRISPR repeat) (Fig. 1c), indicating that their CRISPR-Cas loci are very closely related. Notably, the three IGRs each contains two copies of CRISPR repeat-like sequences, which we denoted as ΨR1 and ΨR2,

respectively (Fig. 1b). The ΨR2 elements are highly conserved in sequence and very similar (~90%) to their CRISPR repeats (Fig. 1c), while ΨR1 sequences markedly diverge from each other. Nevertheless, all ΨR1, ΨR2, and CRISPR repeat sequences conservatively harbor a pair of inverted repeats, indicating their RNA products form a conserved hairpin structure (Fig. 1d), which may be recognized by Csy4 for cleavage[23]. We further analyzed the sequences spacing each pair of ΨR1 and ΨR2 (designated ΨS) and found that they all partially match to a nearby target sequence that is 5′-flanked by a CC dinucleotide (Fig. 1b), which corresponds to the PAM (protospacer adjacent motif) of I-F subtype[24]. Thereby, we inferred these mini CRISPR-like elements to be *creA* genes and these PAM-flanked sequences to be their targets.

Subsequently, within each *cas3-csy1* IGR, we discovered a conserved 33 bp open reading frame (ORF) that is preceded by a Shine-Dalgarno motif (which helps initiating translation by base pairing to the 3′ end of 16 *S* rRNA) (Fig. 1b). Notably, each mini-ORF is oriented divergently from the *creA* gene and locates downstream of the predicted target site of CreA, suggesting their expression may be negatively regulated by CreA. So, we presumed that the I-F CRISPR-Cas loci from *Acinetobacter* species WCHA45, LoGeW2-3, and ANC3789 should contain three homologous *creTA* modules, which we next examined in the type strain *A. baumannii* AYE.

### Reboot CreTA in *A. baumannii* AYE

Due to repeat degeneration, *creTA* has evolved to be highly specific to its genetically-linked CRISPR-Cas, and this specificity could be altered by modifying its repeat elements[22] (see Fig. 2a for the principle of the repeat replacement assay). We noted that ΨR1 and ΨR2 of WCHA45, LoGeW2-3, and ANC3789 *creTA* all share limited nucleotide identity (46–67%) to the CRISPR repeat of *A. baumannii* AYE (Fig. 1c), and inferred that replacing these repeat elements to the CRISPR repeat of AYE should be required to enable these *creTA* modules fitting the AYE CRISPR-Cas system. Therefore, for each *creTA* module, we constructed three modified versions, i.e., ΨR1-replaced, ΨR2-replaced, and both ΨR-replaced derivates (Fig. 2b). Then we transformed AYE cells with plasmids bearing one of these TA modules. LoGeW2-3 *creTA* caused a very low transformation efficiency (compared to the empty vector) no matter the repeat elements were replaced or not (Fig. 2b), suggesting that repeat replacement failed to reboot the antitoxin CreA. In contrast, ANC3789 *creTA* showed no effects on transformation efficiency even when the antitoxin gene *creA* or the nuclease maturing CreA RNA (Csy4) was omitted (Fig. 2b), suggesting ANC3789 *creT* was not deleterious at least to AYE cells. WCHA45 *creTA* caused a 5-log reduction in transformation efficiency prior to repeat replacement, a ~2-log reduction when ΨR1 was replaced by the CRISPR repeat of AYE, and almost no effects on transformation efficiency when ΨR2 or both ΨR elements were replaced (Fig. 2b). We concluded that WCHA45 and LoGeW2-3 *creT* genes were both toxic, while only the *creA* gene of WCHA45 could be rebooted in AYE by repeat replacement (e.g., simply replacing its ΨR2). Therefore, the ΨR2-replaced *creA* was used for subsequent analysis, unless specified.

Using AYE cells transformed by the plasmid expressing WCHA45, LoGeW2-3, or ANC3789 *creA*, we probed CreA RNA by Northern blotting. For each *creA*, we detected very few amounts of mature RNA products from the wild-type antitoxin gene, and increased amounts from those with modified repeat elements (Fig. 2c and Supplementary Fig. 1a), which explained the effects of repeat replacement on CreA activity (Fig. 2b). Next, we identified the exact nucleotide sequence of WCHA45 CreA RNA (ΨR2-replaced) by small RNA sequencing (sRNA-seq) (Fig. 2d). Besides a 20 nt ΨS sequence, the mature CreA RNA carried an 8 nt 5′-handle and a 20 nt 3′-handle, i.e., a typical feature of a I-F crRNA[23]. We supposed that, like the unraveled antitoxic mechanism of *H. hispanica* CreA[19], WCHA45 CreA RNA mimics a canonical crRNA and guides the Csy complex to suppress its cognate toxin gene. Consistently, the antitoxic function of WCHA45 CreA was abrogated in AYE

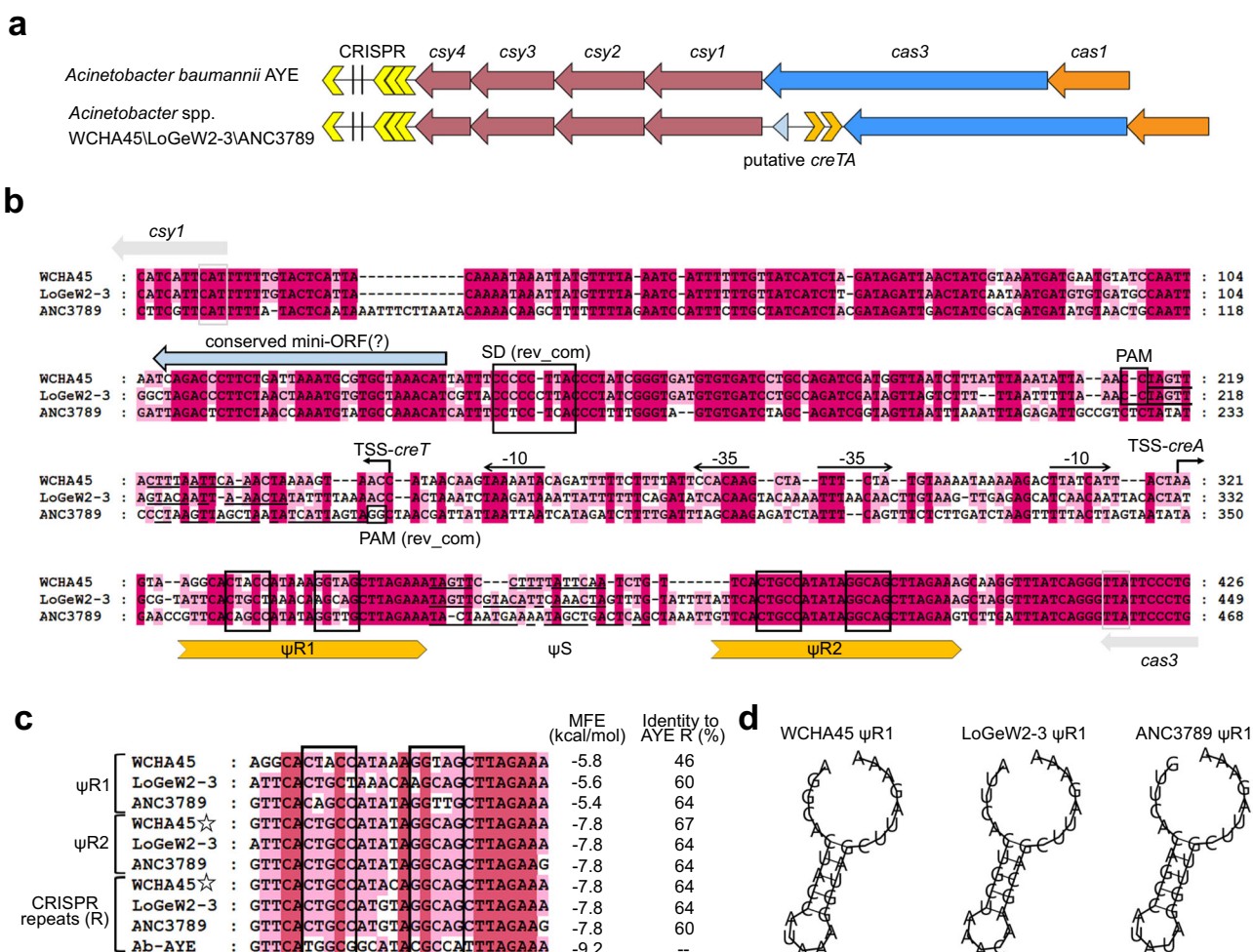

**Fig. 1 | Three putative CreTA modules associating with the type I-F CRISPR-Cas from *Acinetobacter* species. a** Scheme depicting the putative *creTA* module residing within the *cas3-csy1* intergenic region (IGR). The CRISPR-Cas locus of the *A. baumannii* strain AYE which lacks this IGR is given for comparison. **b** Sequence alignment of the three *creTA*-containing IGRs. Each *creA* gene contains a 'spacer' sequence (ΨS) that is sandwiched by two CRISPR repeat-like sequences (ΨR1 and ΨR2). Each ΨR is palindromic and contains a pair of inverted repeats (framed). The identical nucleotides shared between each ΨS and its predicted target are underlined, and the protospacer adjacent motif (PAM) or its reverse complement

(rev_com) is indicated. The *creT* genes carry a seemingly conserved mini open reading frame (ORF) preceded by a Shine-Dalgarno (SD) sequence. Transcription start site (TSS) of *creA* and *creT* was determined by RNA-seq (Fig. 2d) and primer extension (Supplementary Fig. 3), respectively, and their promoter elements (−10 and −35) were correspondingly predicted. **c** Alignment of the repeat sequences of *A. baumannii* CRISPR (R) and *creA* (ΨR1 and ΨR2). For each repeat sequence, its identity to the CRISPR repeat from AYE strain is given. MFE minimal free energy. **d** The hairpin-forming potential of ΨR1 RNA sequences.

cells lacking the Csy complex, but not in cells lacking Cas1 or Cas3 (Supplementary Fig. 2).

### CreA transcriptionally repress its cognate *creT*

Then we characterized the promoter of WCHA45 *creT* (P$_{creT}$). From the sRNA-seq data, we could retrieve very few *creT* reads (possibly due to the repression effect of CreA), so we further performed Northern blotting and detected CreT RNA products of varying sizes (Supplementary Fig. 1b), suggesting multiple transcription start/terminate sites or putative processing events. Then we engineered a variant of *creT*, of which the mini-ORF was replaced to the *gfp* (green fluorescence protein) reporter gene (depicted in Supplementary Fig. 3 and Fig. 3b), to perform primer extension analysis (Supplementary Fig. 3). In AYE cells, we determined the transcription start site (TSS; upstream of the target site of CreA) and predicted the −35 and −10 elements (indicated in Fig. 3a). As expected, fluorescence produced from this P$_{creT}$-controlled *gfp* disappeared when these elements were omitted, or when the −10 element was mutated (Fig. 3b).

Then we added the *creA* gene (ΨR2-replaced) to this P$_{creT}$-*gfp* construct, which markedly suppressed fluorescence production

(Fig. 3b). Notably, this suppression was subverted when the gene encoding Csy4 (responsible for CreA RNA maturation) was deleted, supporting the regulatory role of mature CreA. By subjecting the same batch of cell samples to fluorescence measurement and RNA abundance analysis, we further showed that AYE cells containing only the P$_{creT}$-controlled *gfp* produced transcripts ~7.25-fold those from cells also containing the *creA* gene, which was comparable to the fold change in fluorescence (~7.87-fold) (Fig. 3c). Therefore, CreA should regulate *creT* expression at the transcription level. Note that, though CreA complements also with the RNA transcripts of *creT*, the CreA-Csy complex theoretically cannot bind to these transcripts because type I CRISPR effectors only recognize DNA targets (where the PAM motif plays a critical role during target recognition).

### CreA does not have a canonical seed in recognizing its target

To confirm CreA acts as an RNA guide to repress P$_{creT}$, we analyzed the necessity of the complementarity between CreA and its target (downstream of P$_{creT}$). The ΨS portion of CreA has 20 nucleotides, 17 of which base pair to the target DNA (except the 6th, 11th, and 18th) (Fig. 3a). We separately mutated each of these 17 nucleotides to

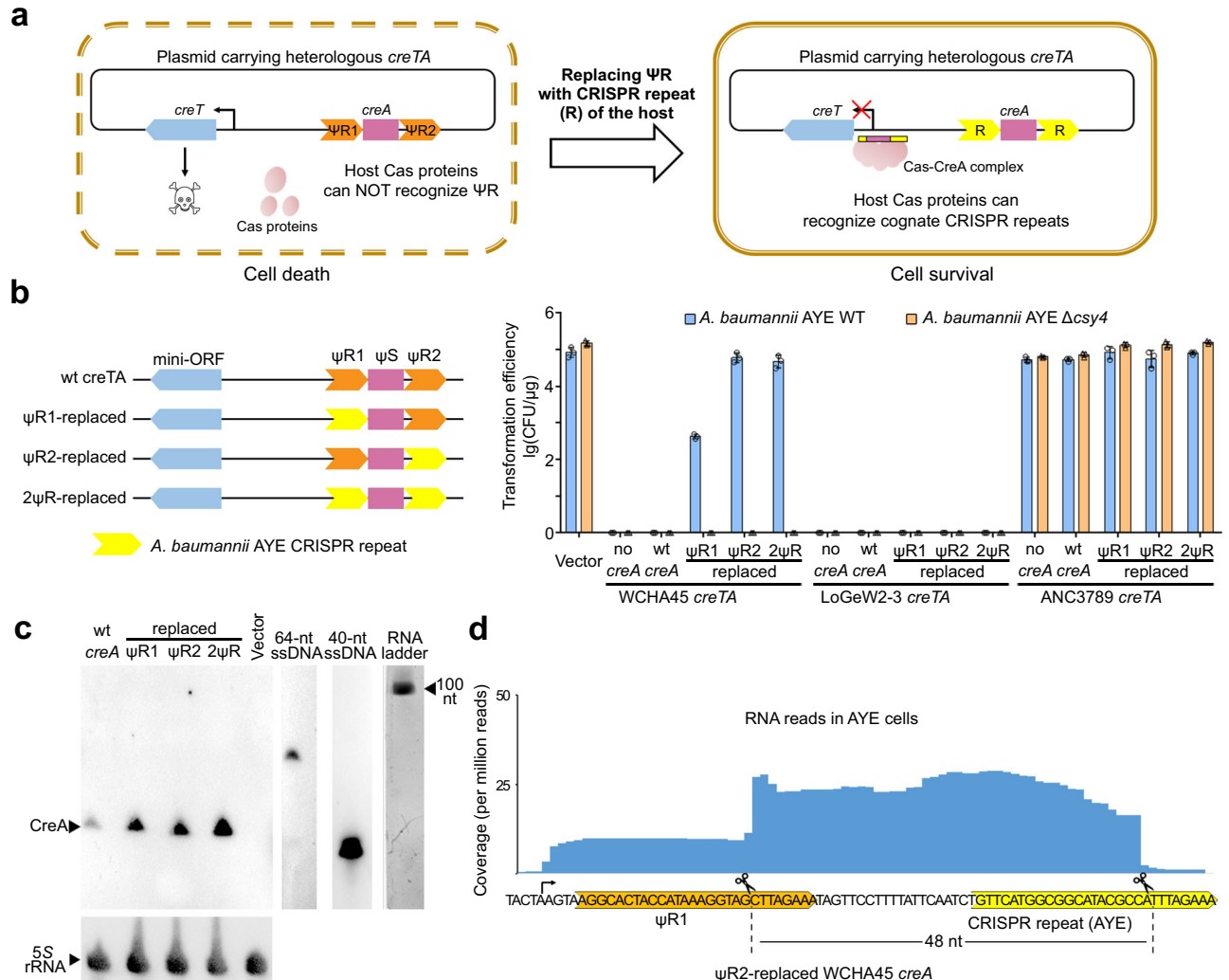

**Fig. 2 | Repeat replacement to reboot heterologous CreTA in *A. baumannii* AYE.**
**a** Scheme illustrating the principle of repeat replacement. The host Cas proteins cannot be guided by a heterologous *creA* to repress toxin expression unless the degenerated repeats (ΨR1 and ΨR2, in orange) are replaced by the cognate CRISPR repeats (R, in yellow) of Cas proteins. **b** Transformation efficiency of *A. baumannii* AYE cells by plasmids carrying a wild-type (wt) *creTA* or its derivates where one or both ΨR elements were replaced by the CRISPR repeat of AYE. CFU, colony

formation unit. Data are presented as mean value ± s.d. (*n* = 3 biological replicates). **c** Northern blotting of CreA molecules produced from a wt or a repeat-replaced WCHA45 *creA* gene. Plasmids bearing only the *creA* gene (without *creT*) were introduced into AYE cells for this assay. All data were acquired from at least three independent replicates with similar results. **d** Identify the sequence of the WCHA45 CreA RNA (ΨR2-replaced) by small RNA sequencing. Scissors indicate the cleavage sites of Csy4. Source data are provided as a Source Data file.

eliminate their base pairing potential to the target DNA, and transformed AYE cells with plasmids carrying these mutated *creTA* (Fig. 3d). Usually, the first 8 or 11 base pairings adjacent to the PAM motif provide a seed during the formation of crRNA-target DNA duplex and are more important for target recognition[25,26]. However, we found that, when any of the nucleotides 2–5, 7–9, and 13–17 was mutated, CreA failed to suppress the toxin, which caused a 4-log reduction (compared to the empty vector) in the efficiency of transforming AYE cells (Fig. 3d). In contrast, the antitoxic function of CreA was not influenced when the 10th, 12th, 19th, or 20th base pairing was disrupted. Rather unexpected, when the 1st base pairing (next to PAM) was disrupted, CreA appeared to be only partially inactivated and a reduction less than 2-log (compared to the empty vector) was observed in transformation efficiency (Fig. 3d). Therefore, it seemed that CreA does not follow the canonical seed rule when binding to its target DNA.

We further selected three base pairings (the 5th, 7th, and 15th) to perform complementary mutation analysis (Fig. 3e). CreA failed to suppress the toxin (causing a markedly reduced transformation efficiency) when itself or its target DNA was mutated at any of these three

nucleotide positions. However, when CreA and the target DNA were simultaneously mutated with their complementarity well maintained, CreA kept its antitoxic function and the engineered plasmids transformed AYE cells with an efficiency equivalent to the empty vector (Fig. 3e). These data substantially support that CreA guides toxin repression based on its partial complementarity to the target site downstream of P$_{creT}$.

## CreT is an RNA toxin with strong translation-initiating signals
Next, we investigated the toxicity mechanism of WCHA45 CreT. We first replaced P$_{creT}$ to a synthetic *tac* promoter (P$_{tac}$) and then constructed a series of differently truncated *creT* genes to determine its key elements (Fig. 4a). The data showed that, under the control of P$_{tac}$, the conserved mini-ORF and the predicted SD motif were sufficient to cause toxicity in AYE cells. We further showed that toxicity almost disappeared when the putative SD motif was truncated (Fig. 4a) or mutated (Supplementary Fig. 4a). Notably, replacing this motif to a canonical *E. coli* SD sequence[27] did not influence the toxicity (Supplementary Fig. 4a). Besides, using the *gfp* reporter gene, we observed

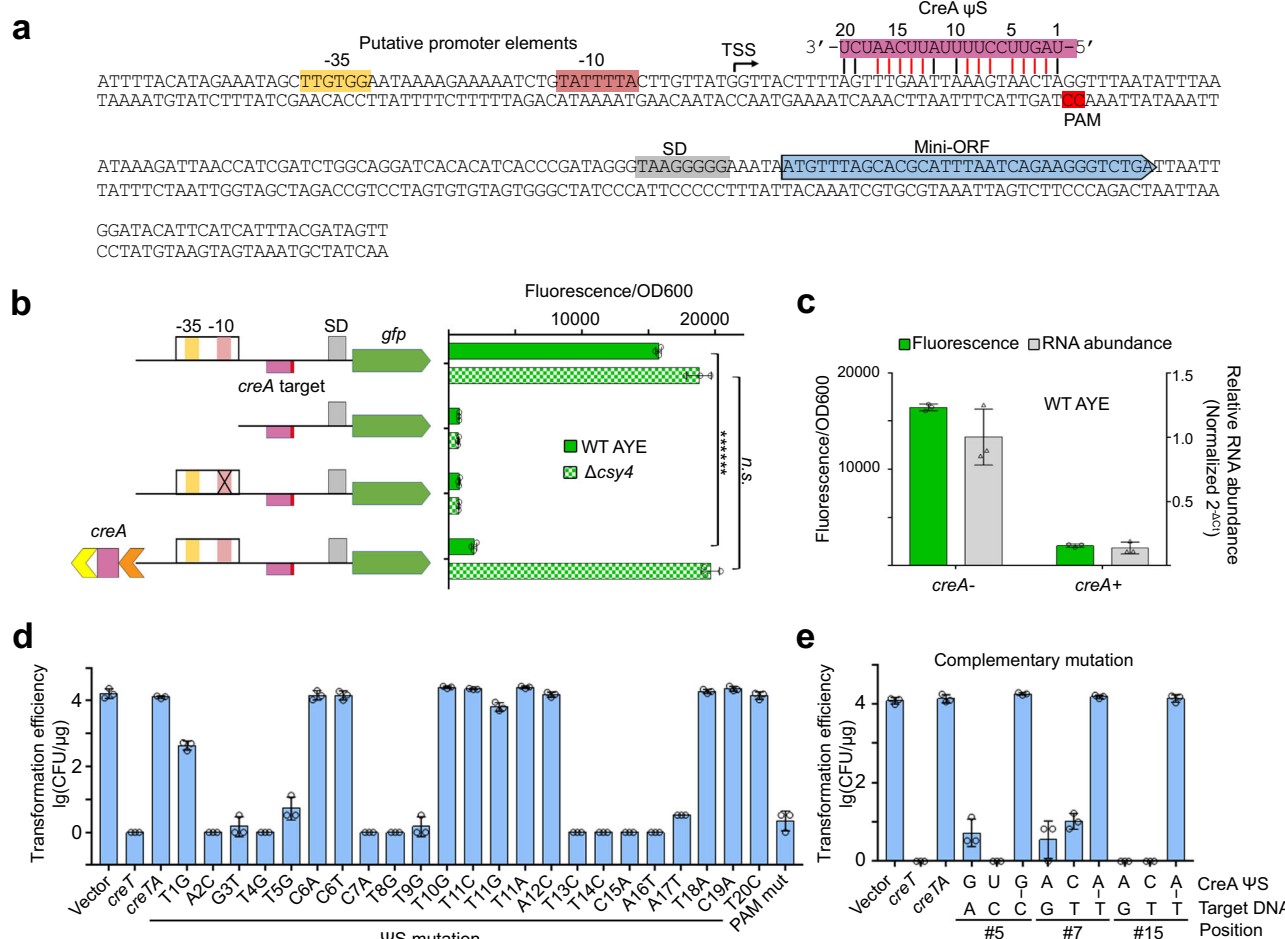

**Fig. 3 | WCHA45 CreA transcriptionally represses its cognate *creT*. a** The target site of CreA. Base pairings between CreA and its target are indicated by "|" with the red ones proved to be critical for its antitoxin function (see **d**). **b** Validation of the predicted promoter elements of *creT* and its repression by CreA using the *gfp* reporter gene. Note that the ΨR2-replaced CreA was employed for this and following assays. Data are presented as mean value ± s.d. (*n* = 3 biological replicates). Two-tailed Student's *t* test [*n.s.*, not significant (*P* > 0.05); *******P* < 0.000001. *P* = 1.46894E−07, 0.274299515 (top to bottom)]. **c** Assess the repressing effects of CreA on the protein and RNA levels of *gfp* by fluorescence and qPCR assays, respectively. Data are presented as mean value ± s.d. (*n* = 3 biological replicates). **d** Point mutations within CreA ΨS sequence and their effects on its antitoxin function. AYE cells were transformed by plasmids carrying *creT* and *creA* derivates (all ΨR2-replaced). A PAM mutant was also analyzed. **e** Complementary mutation analysis between CreA ΨS and its target. Data are presented as mean value ± s.d. (*n* = 3 biological replicates). Source data are provided as a Source Data file.

that the predicted SD motif of *creT* greatly improved the translation efficiency of GFP protein (compared to an SD-minus control) (Supplementary Fig. 4b). These data collectively support our prediction of the SD motif and suggest that an efficient translation initiation process should be required for toxicity.

Intriguingly, we found that the stop codon of the mini-ORF was dispensable for toxicity (Fig. 4a), and when this mini-ORF was fused to a *gfp* gene, toxicity was not influenced (Supplementary Fig. 5b). These data led us to doubt whether WCHA45 *creT* results in toxicity via its potential protein product. So, we subjected each codon (except start and stop codons) of the mini-ORF (not fused with *gfp*) to synonymous mutation, which in principle did not alter the encoded amino acids. Remarkably, when the 2nd (TTT), 3rd (AGC), or 5th (CAT) codon was changed to a synonymous alternative, the *creT* gene became non-toxic to AYE cells (Supplementary Fig. 5a), indicating that the nucleotide sequence of *creT* RNA rather than its potential encoded amino acids dictate the toxicity. Because synonymous codons are used with different frequencies in bacteria and replacing a rich codon to a rare one may alter the amounts of protein products, we further mutated the 3rd codon (AGC) to any of the other five serine codons, which are utilized more frequently (TCT, TCA, and AGT) or less frequently (TCG and TCC) than AGC in AYE, and fused each of these mutated mini-ORFs

to the *gfp* gene (Supplementary Fig. 5b). Notably, only when the wild-type codon was employed did we observe toxicity in AYE cells. In contrast, fluorescence production was not abrogated by any of these alternative codons, and especially, in the case of TCT (the most frequently utilized serine codon in AYE), the mutated *creT-gfp* fusion showed the strongest fluorescence intensity (Supplementary Fig. 5b). Therefore, we conclude that the toxicity of *creT* does not rely on its potential protein product, though the mini-ORF appears to be rather conserved. In fact, the hitherto characterized archaeal *creT* genes also conservatively carry a mini-ORF, but they were experimentally demonstrated to similarly act as a toxic RNA (by sequestering rare tRNA species)[19,21].

**CreT toxicity relies on both a hairpin structure and a start codon**
Then we asked how the RNA of *creT* causes cellular toxicity. We found that CreT RNA could form a hairpin structure, with the start codon AUG locates in the center of a 5 bp stem (Fig. 4b). We numbered the nucleotides with respect to the start codon and first mutated U4 and G12, which form a wobble base pair joining the stem and the loop. Toxicity disappeared when U4 was mutated to A or G that does not base pair to G12, while persisted when mutated to C that forms a Watson-Crick base pair with G12 (Fig. 4c). Similarly, toxicity

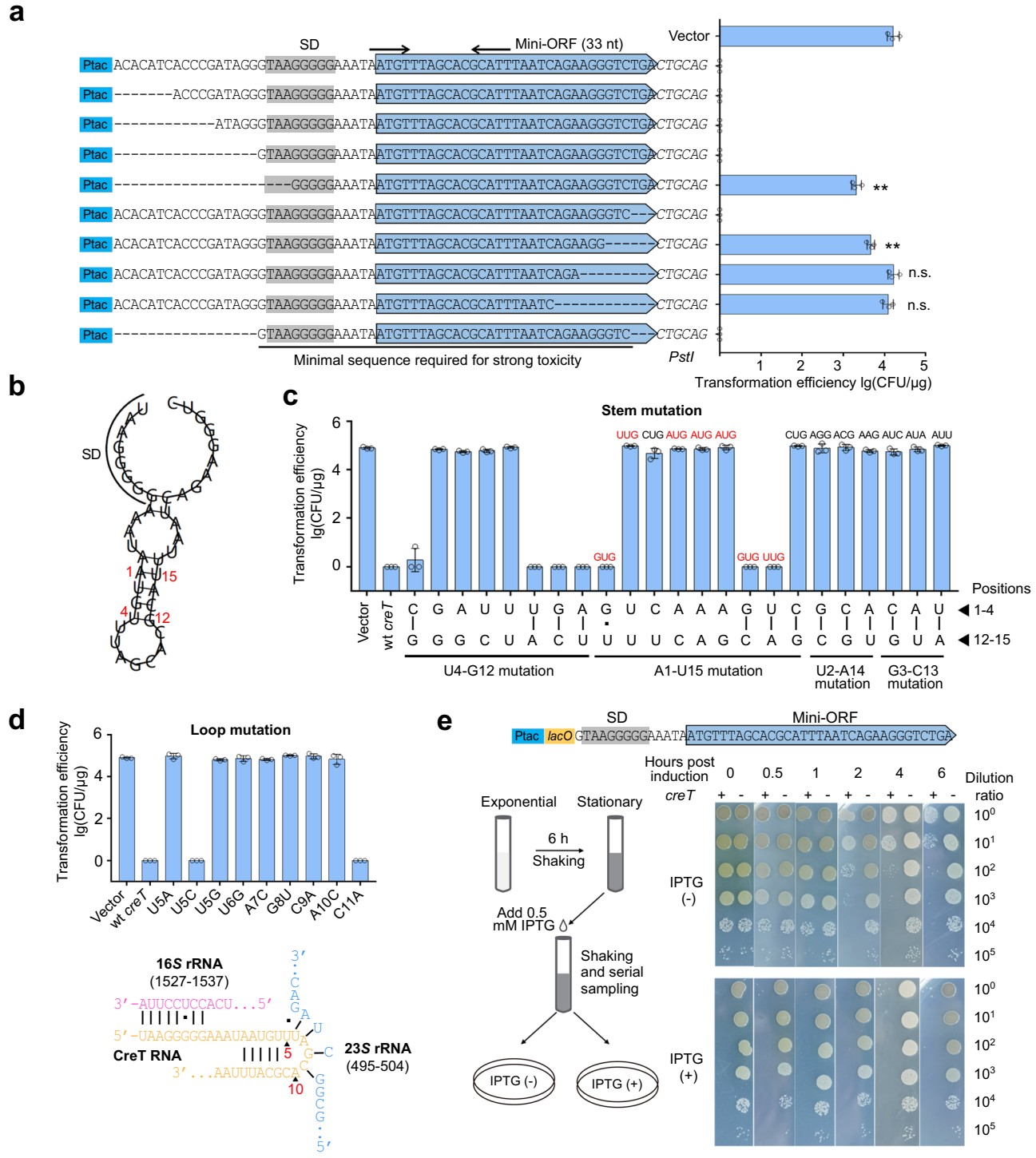

**Fig. 4 | WCHA45 CreT is a bactericidal RNA that relies on translation initiation signals and a hairpin structure. a** Truncation assay to determine the minimal sequence of *creT* that is required for toxicity. A synthetic *tac* promoter (P*tac*) was used to drive *creT* transcription. Plasmids expressing CreT variants were used to transform AYE cells. Data are presented as mean value ± s.d. (*n* = 3 biological replicates). Two-tailed Student's *t* test [*n.s.*, not significant (*P* > 0.05); **P* < 0.01. *P* = 0.001229433, 0.005484642, 0.960215311, 0.326149613 (top to bottom)]. **b** The predicted hairpin structure of CreT RNA. Nucleotide positions are indicated with respect to the start codon AUG. **c** Complementary mutation of the stem nucleotides. Plasmids expressing mutated CreT were used to transform AYE cells. For some mutants, their nucleotides at the position of the original start codon are given, with those capable of initiating translation (AUG, GUG, and UUG) shown in red. Data are presented as mean value ± s.d. (*n* = 3 biological replicates. **d** Mutation analysis of the loop nucleotides. The base pairing potential of loop nucleotides to 23 *S* rRNA is depicted. Data are presented as mean value ± s.d. (*n* = 3 biological replicates. **e** Assessing the toxicity (bacteriostatic or bactericidal) of CreT by dilution plating after its induction. For this assay, a *lacO* operator was inserted between P*tac* and SD, and the expressing plasmid also carried a *lacI* gene. Source data are provided as a Source Data file.

disappeared when G12 was mutated C or U (not complementary to U4), while persisted when mutated to A (complementary to U4). Then we complementarily mutated these two nucleotides to a G-C or a A-U base pair, and found that toxicity persisted (Fig. 4c). These data demonstrated that the complementarity rather than the identity of the 4th and the 12th nucleotides was critical for the toxicity of CreT RNA.

Then we analyzed the complementarity between A1 and U15. Similarly, when A1 or U15 was mutated to disrupt their complementarity, toxicity was no longer observed in AYE cells (transformation efficiency was equivalent to the empty vector) (Fig. 4c). Notably, when A1 was altered to G, which forms a wobble base pair with U15, CreT RNA remained to be toxic and markedly reduced transformation efficiency. Unexpectedly, toxicity was observed when A1 and U15 were complementarily mutated to a G-C or a U-A base pair, but not when mutated to a C-G pair (Fig. 4c). Because AUG, GUG, and UUG can all act as a start codon to initiate translation, we inferred that both the hairpin structure and the start codon be indispensable for toxicity. Consistently, when U2 and A14 were mutated to a G-C, C-G, or A-U base pair, or when G3 and C13 were mutated to a C-G, A-U, or U-A pair, toxicity was no longer observed (Fig. 4c). Using an IPTG-inducible promoter, we further showed that the CreT RNA with an efficient start codon (AUG or GUG) markedly impaired the growth of AYE cells in medium containing 0.2 mM IPTG (with the most efficient AUG showing a stronger effect), while CreT with the inefficient start codon UUG could impair cell growth only in medium containing 0.4 mM or more IPTG (note that the hairpin structure was maintained in all these mutants) (Supplementary Fig. 6). It appeared that a more efficient start codon could lead to higher-level toxicity.

**CreT is a bactericidal small RNA likely targeting ribosomal RNAs**
Then we analyzed the loop nucleotides by scanning mutation. CreT became non-toxic after any of the 7 loop nucleotides (except C11) was subjected to a transversion substitution (i.e., purine to pyrimidine, or the converse) (Fig. 4d), indicating these nucleotides play critical roles in CreT activity. Interestingly, we found that nucleotides 5−9 could complement with 23 S rRNA (Fig. 4d), implying a possible interaction between CreT and 23 S rRNA. We also noted that U5 could form a wobble base pair to G502 of 23 S rRNA, thus further mutated this loop nucleotide to C or A. Notably, only mutation to C (forming a Watson-Crick base pair to G502) retained the toxicity of CreT (Fig. 4d). Therefore, we surmise that, in addition to interacting with 16 S rRNA via the SD motif, CreT might also interacts with 23 S rRNA via the loop.

To assess the toxicity effect (bactericidal or bacteriostatic) of WCHA45 CreT, we induced CreT expression after the AYE cell culture reached stationary phase, and then plated the culture samples onto inducing or non-inducing medium after serial dilutions (Fig. 4e). As expected, cells did not grow on inducing plates, but notably on the non-inducing plates, CreT almost caused a 2-log reduction in colony formation units (CFU) 2 h post induction, and a 4-log reduction 6 h post induction, indicating 99.99% individual cells were at last killed. Therefore, we conclude that WCHA45 CreT is a bactericidal RNA toxin.

**CreTA improves the antimicrobial effects of a self-targeting CRISPR**
Because CreTA safeguards the CRISPR effector in bacterial population[19], we expected CreTA could improve the performance of CRISPR antimicrobials. We tested this potential first by targeting the MDR *A. baumannii* AYE cells that harbor an endogenous CRISPR-Cas (Fig. 1a). For CRISPR design, we selected the putative gentamicin-resistant gene *aac3* (or *aacC1*; GenBank ID: ABAYE3573) as the target[28]. By in-frame deleting *aac3* and then complementing a plasmid copy of this gene, we confirmed that *aac3* determines the gentamicin resistance of AYE (Supplementary Fig. 7). Then we transformed WT AYE cells separately with pAAC3 (bearing a mini-CRISPR with a spacer targeting *aac3*) and pAAC3-TA (also contains the ΨR2-replaced WCHA45

*creTA* module). On the medium containing potassium tellurite (Pt, selecting for all transformants), pAAC3 and pAAC3-TA reduced transformation by 4-fold and 61-fold, respectively, compared to the empty vector (Fig. 5a). That means, CreTA elevated the bacteria-killing efficiency of CRISPR from 75.08 to 98.37%. On the medium containing Pt and gentamicin (selecting for gentamicin-resistant transformants), pAAC3 and pAAC3-TA reduced colonies by ~471-fold and ~4-log, respectively, compared to the empty vector (Fig. 5a). We calculated that, among the survivors from CRISPR killing, the ratio of cells that lost gentamicin resistance was 99.39% for pAAC3 and 99.86% for pAAC3-TA, suggesting CreTA also improved the AR-curing effects. IS insertion into *cas* genes represents one of the various resistant mechanisms to CRIPSR antimicrobials, and could be readily detected by colony PCR. So, we amplified the *csy* operon of the gentamicin-resistant survivors, and found that it was interrupted by an IS element (genomic position: 1271633-1272960) in 30% (9 out of 30) survivors from pAAC killing, but not in any survivors from pAAC3-TA killing (Supplementary Fig. 8), which exemplifies the protective effect of CreTA on CRISPR antimicrobial.

To rule out the possibility that the observed improving effect of CreTA on CRISPR antimicrobial derived (partially) from potential CRISPR-independent toxicity of the plasmid-born TA module, we inserted *creTA* into the AYE chromosome, within (TA-in) or outside (TA-out) the *cas* operon, and transformed these mutant cells with pAAC3 (Fig. 5b). On each medium, the empty vector transformed TA-in or TA-out cells with an efficiency comparable to its efficiency in transforming WT AYE cells. By contrast, pAAC3 reduced WT, TA-in, and TA-out colonies on the Pt-containing medium by a factor of 4.01, 14.18, and 9.47, respectively, and on the gentamicin-containing (plus Pt) medium by a factor of 471, 4008, and 10100, respectively. Compared to the data in WT AYE cells, we concluded that, no matter within or outside the *cas* operon, *creTA* did elevate the efficiency of CRISPR killing (from 75.08% to 92.95% (TA-in) or 89.44% (TA-out)), and in the survivors, improve the AR-curing efficiency (from 99.39% to 99.65% (TA-in) or 99.89% (TA-out)). Therefore, by making the bacterial cells addicted to CRISPR effectors, CreTA improved both the bacteria-killing and the AR-curing effects of the CRISPR antimicrobial.

We also tested the effect of CreTA when programming CRISPR to cure a plasmid-born *aac3* gene (Supplementary Fig. 9). Compared to the transformed colonies on non-selective medium (containing no gentamycin), pAAC3 and pAAC3-TA both showed highly reduced transformation efficiency (by a factor of 6424 and 2959, respectively) on medium containing 8 μg/ml gentamicin, indicating >99.9% of pAAC3 and pAAC3-TA transformants could not grow under this selection pressure. Interestingly, on medium containing 4 μg/ml gentamicin, the colonies of pAAC3 and pAAC3-TA were reduced by 20% ($P = 0.34$) and 90% ($P = 1.68e−04$), respectively (Supplementary Fig. 9), indicating pAAC3-TA transformants were more sensitive to a low concentration of antibiotic compared to pAAC3 transformants. Therefore, in the presence of CreTA, CRISPR antimicrobials could cure antibiotic resistance more completely.

**AssociaTe TA and CRISPR-Cas to kill (ATTACK) a clinical CRAB**
Next, we attempted to combine CreTA and CRISPR-Cas to kill MDR clinical isolates of *A. baumannii* (Fig. 6a). First, we tested the feasibility of the combination of WCHA45 CreTA (ΨR2-replaced) and AYE CRISPR-Cas in clinical isolates. We constructed pCas (carrying the AYE *cas3-csy* operon), pCasT (carrying the *cas3-csy* operon and *creT*), and pCasTA (carrying the *cas3-csy* and *creTA* operons), and transformed 18 clinical MDR isolates (see Supplementary Data. 1 for more strain information) and the type strains ATCC 17978 and ATCC 19606 of *A. baumannii* (Supplementary Fig. 10). Compared to pCas, addition of *creT* (pCasT) greatly reduced the transformation efficiency of each strain, which was then rescued by further adding *creA* to the plasmid (pCasTA). It was suggested that the WCHA45 CreT toxin and the

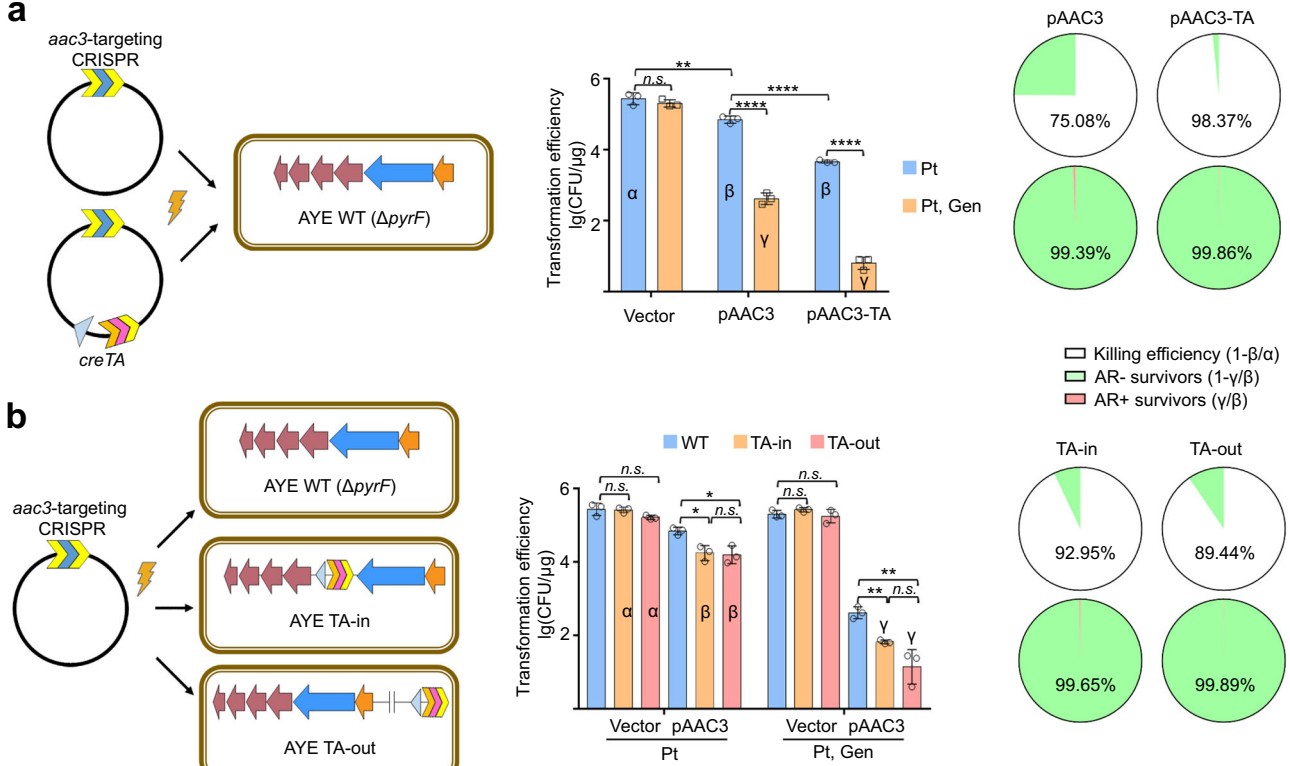

**Fig. 5 | An *aac3*-targeting CRISPR killed *A. baumannii* AYE cells more efficiently in the presence of CreTA. a** Kill AYE cells using an *aac3*-targeting CRISPR alone (pAAC3) or together with a ΨR-replaced WCHA45 *creTA* module (pAAC3-TA). Data are presented as mean value ± s.d. (*n* = 3 biological replicates). Two-tailed Student's *t* test [*n.s.*, not significant (*P* > 0.05); **\*\****P* < 0.01; **\*\*\*\****P* < 0.0001. *P* = 0.314515862, 0.006575012, 3.55829E−05, 4.80705E−05, 1.00223E−05 (left to right)]. **b** Using pAAC3 to kill AYE cells containing a *creTA* module that was inserted within (TA-in) or outside (TA-out) the *cas* operon (between *cas3* and *csy1*). CRISPR was driven by the native leader of AYE CRISPR array. Plasmids expressing the genetic anti-microbials were introduced into AYE cells by electro-transformation. Potassium tellurite (Pt) selected for all transformants (the vector carried a potassium tellurite-resistant gene), while the combination of Pt and gentamicin (Gen) select for the antibiotic resistant ones. CRISPR-killing efficiency and the ratio of antibiotic resistant (AR+) or sensitive (AR−) survivors were calculated. Data are presented as mean value ± s.d. (*n* = 3 biological replicates). Two-tailed Student's *t* test [*n.s.*, not significant (*P* > 0.05); *\*P* < 0.05; **\*\****P* < 0.01. *P* = 0.891231406, 0.098091559, 0.010477841, 0.012724006, 0.808148096, 0.159770955, 0.676328189, 0.001258506, 0.006747047, 0.067773026 (left to right)]. Source data are provided as a Source Data file.

antitoxic complex, composed of WCHA45 CreA and AYE Csy proteins, were both functional in all these clinical isolates and type strains.

According to their genomic information, we found that these clinical isolates all carry the most common carbapenemase gene *oxa23*, which confers on pathogens resistance to carbapenem antibiotics, like meropenem[29]. So, we designed a mini-CRISPR with its spacer targeting *oxa23*, and placed it under the control of the synthetic promoter J23114 and a *lacO* operator so that CRISPR targeting could be inducible by IPTG (Fig. 6a). Then we constructed a conjugative plasmid carrying this CRISPR, *cas3-csy* operon, *creTA*, a *lacI* gene and a potassium tellurite-resistance gene, and named it pATTACK. Using the *E. coli* S17-1 donor cells, we delivered pATTACK into cells of a clinical *A. baumannii* isolate (Aba14 in Supplementary Fig. 10) via the filter mating assay, and plated the conjugants onto different media (Fig. 6b). Pt (potassium tellurite, selecting for all cells containing pATTACK) and kanamycin (to kill donor cells) together selected for all conjugants, Pt, kanamycin, and IPTG selected for conjugants that experienced and survived from CRISPR killing, while Pt, meropenem (to kill donors and AR-cured recipients), and IPTG together selected for meropenem resistant survivors. Using a non-targeting CRISPR, we obtained equivalent numbers of conjugants in the presence or absence of *creTA* on each medium (Fig. 6c), which reaffirmed that creTA per se did not cause antimicrobial effects. Upon induction, the *oxa23*-targeting CRISPR reduced the number of conjugants by a factor of 46.24 or 95.85 in the absence or presence of *creTA* (Fig. 6c), indicating that CreTA

improved the efficiency of CRISPR antimicrobial (literally from 97.84 to 98.96% with a *P* value of 2.61e−07). Among the survivors, the ratio of cells losing meropenem resistance was calculated to be 91.38% and 99.64% in the absence and presence of *creTA*, respectively. Therefore, CreTA markedly improved the AR-curing effect. In conclusion, the ATTACK strategy eliminated and cured MDR pathogens more efficiently than the conventional CRISPR antimicrobial strategy.

## Discussion

Characteristically of immune systems, CRISPR-Cas not only confers on bacterial cells the adaptive immunity, but also imparts non-negligible fitness costs on the host, which are believed to come, primarily, from autoimmune risks and expelling of beneficial genes (e.g., antibiotic resistance genes)[30–33]. Therefore, bacteria have evolved intrinsic mechanisms to inactivate or compromise CRISPR immunity to gain a tradeoff between its benefits and downsides. For example, *cas* genes can be destructed by various genetic variations, like spontaneous nucleotide mutations, chromosomal rearrangements, MGE transpositions, etc[3,4,16,34]. Besides, Cas proteins can be inactivated by the widely-distributed Acr proteins[17,18]. These genetic variations/elements will bring about selective benefits in the presence of a self-targeting CRISPR spacer, especially when the target gene is beneficial to the bacterial cell. Therefore, CRISPR antimicrobials, which employ a spacer usually targeting AR or other genes potentially beneficial for bacteria, will inevitably be partly frustrated by these factors.

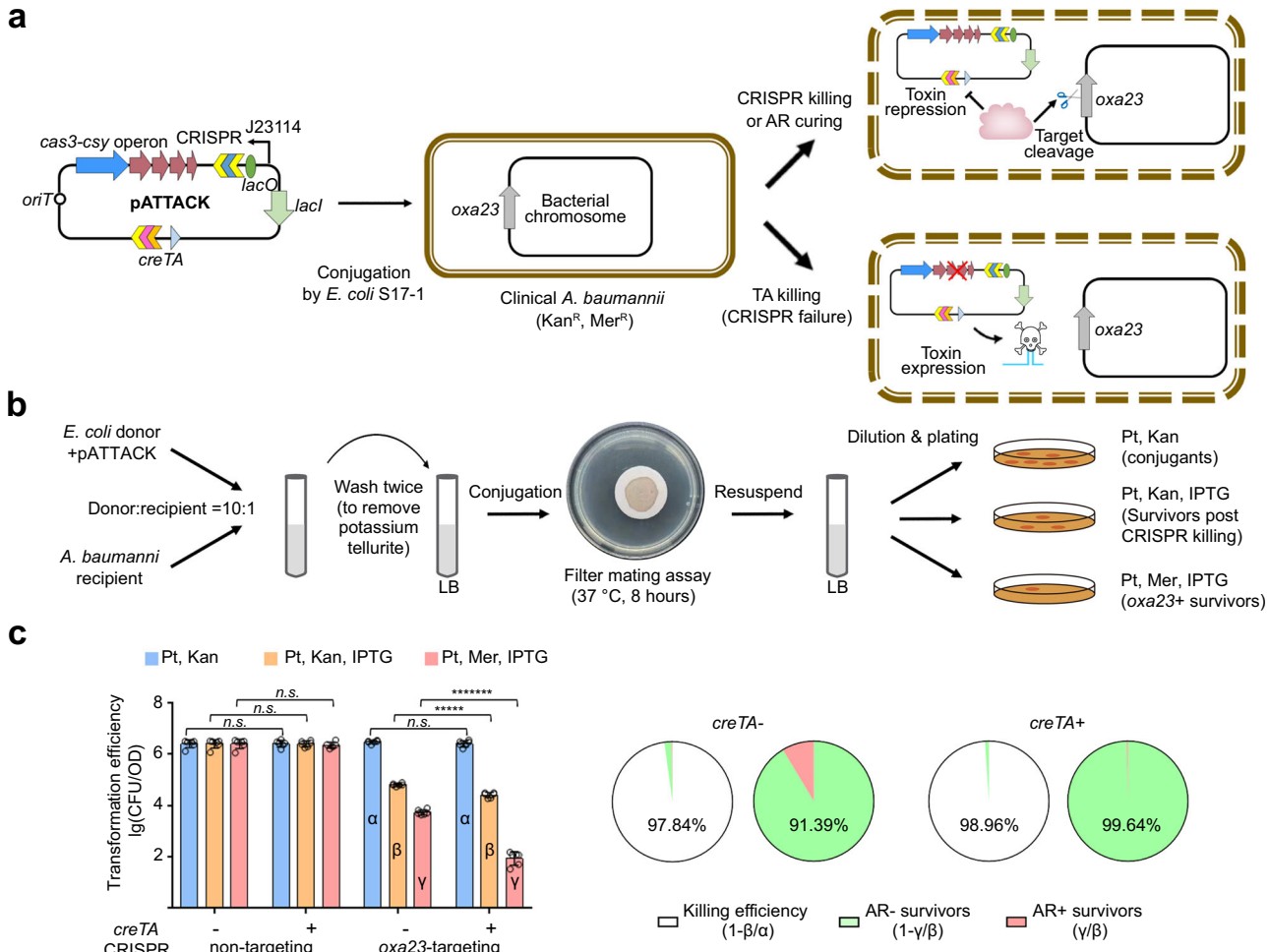

**Fig. 6 | CreTA improved the killing and curing effects of CRISPR antimicrobials against an MDR *A. baumannii* clinical isolate. a** Design of pATTACK. Note that the *oxa23*-targeting CRISPR was controlled by a J23114 promoter with a *lacO* operator (repressed by LacI and induced by IPTG). *oriT*, the origin of conjugation transfer. **b** The flow chart of the conjugation assay to kill a clinical *A. baumannii* using pATTACK. **c** The bacteria killing and antibiotic resistance (AR) curing effects of CRISPR antimicrobial with or without *creTA*. Potassium tellurite (Pt) and kanamycin (Kan, to kill *E. coli* donors) selected for all conjugants, Pt, Kan, and IPTG together selected for conjugants that survived from CRISPR targeting, while Pt,

meropenem (Mer, kill *E. coli* donors and re-sensitized *A. baumannii* recipients), and IPTG together selected for survivors retaining meropenem resistance. CRISPR-killing efficiency and the ratio of meropenem resistant (AR+) or sensitive (AR−) survivors were calculated. Data are presented as mean value ± s.d. (*n* = 6 biological replicates). Two-tailed Student's *t* test [*n.s.*, not significant (*P* > 0.05); *****P* < 0.00001; *******P* < 0.0000001. *P* = 0.960431982, 0.970156895, 0.633788636, 0.255633996, 6.63211E−06, 2.12205E−08 (left to right)]. Source data are provided as a Source Data file.

To combat bacterial resistance to CRISPR antimicrobials, Yosef et al. developed a delicate strategy which combines the use of engineered lytic phages[10]. In this case, the target sequences of the CRISPR antimicrobial (i.e., DNA fragments from the targeted AR genes) were inserted into the genome of the lytic phage, which conferred phage immunity on cells containing active CRISPR antimicrobials (and losing AR genes). Thereby, this phage could be used to specifically eradicate the antibiotic resistant pathogens that had evolved resistance to CRISPR antimicrobials. In theory, this strategy requires the labor-intensive screening of a lytic phage that specifically infects the target pathogen and customized engineering of this phage according to the target gene.

In this study, we propose an alternative strategy based on our recent finding of a toxin-antitoxin module, CreTA[19]. CreTA makes the bacterial cells addicted to CRISPR immunity, and its finding uncovered the selfish feature of CRISPR-Cas, which likely has evolved to mitigate the fitness costs of CRISPR. By exploiting such TA modules, we developed the ATTACK antimicrobial strategy to associate TA and CRISPR-Cas to kill MDR pathogens (Fig. 6). Like conventional CRISPR antimicrobials, CRISPR-Cas is programed to cleave the AR gene and

achieve specific eradication or re-sensitization of MDR pathogens. When a small fraction of pathogens survive by inactivating the CRISPR effector, CreTA produces bactericidal small RNAs to kill the survivors, which proved to improve the final bacteria-killing and AR-curing effects (Figs. 5 and 6; Supplementary Fig. 9). Notably, the data from Supplementary Fig. 9 suggested that CRISPR-Cas activity can be compromised in the absence of CreTA, which lead to incomplete AR curing and survival under lower-level antibiotic selection. Therefore, we propose that CreTA not only improves the genetic integrity of CRISPR-Cas, but possibly also ensures its substantial expression and/or targeting activity in the cell population.

ATTACK is akin to the lytic phage strategy[10] in that they both punish the survivors that resists CRISPR-Cas and retains AR genes. Notably, we observed that a CreTA module can produce CRISPR-regulated toxicity in all tested *A. baumannii* strains/isolates (Supplementary Fig. 10). Combining that its toxicity does not rely on the target gene, ATTACK seems to be more versatile compared to the lytic phage strategy. Nevertheless, the lytic phage strategy also has an obvious advantage, i.e., elimination of pathogens that CRISPR antimicrobials do not reach. Therefore, we expect combining these two strategies will

bring about more powerful antimicrobials. Specifically, to combine these two strategies to combat CRAB, characterization and engineering of *A. baumannii* phages are currently still required (also important for the in vivo delivery and application of ATTACK antimicrobials).

Our finding of this bacterial CreTA reinforces our previous proposal that CRISPR-regulated TA modules are more common than we previously reported[19], which suggests a great potential to develop stable CRISPR antimicrobials. However, the small size and poor sequence conservation of CreT and CreA RNAs have impeded their systemic discovery[21]. For example, we could only find two homologs for WCHA45 CreTA in the NCBI nucleotide database based on their sequence similarity (Fig. 1b). Yet, interestingly, when we searched the sequences surrounding the *csy1* genes that are most related in amino acid sequence to WCHA45 *csy1*, we did manage to find a dozen CreTA homologs (Supplementary Data. 2), which share the conserved 33 bp mini-ORF (of *creT*) and similar ΨR sequences (of *creA*). Therefore, we propose that closely related CRISPR-Cas systems tend to share homologous CreTA modules, which will contribute to the prediction of novel CreTA and to the development of stable CRISPR antimicrobials.

Beside the two archaeal CreTA modules from *H. hispanica*[19] and *Halobacterium hubeiense*[21], the WCHA45 module seems to be the first characterized bacterial CreTA. This bacterial CreTA differs from the two archaeal analogs in several aspects. First, WCHA45 CreT is a bactericidal small RNA (Fig. 4e), while the two archaeal CreT toxins proved to be bacteriostatic[19]. Second, WCHA45 CreA does not follow the canonical seed rule when recognizing its target site (Fig. 3d), while a canonical 11 bp seed region (next to PAM) was observed for both archaeal CreA RNAs[19,21]. Third, the protecting specificity of CreTA is mainly decided by the more conserved ΨR2 element in the bacterial case (Fig. 2), while primarily decided by the divergently-evolved ΨR1 element in the archaeal cases[22].

At last, though our data indicate CreT probably disrupts the translation process, the exact target has not been experimentally identified. We tried to screen an AYE mutant resistant to CreT toxicity, but unfortunately, we could not obtain a mutant with stable resistance. More in vivo and in vitro experiments are required to fully understand this interesting bactericidal RNA toxin.

In conclusion, our data highlighted the diversity of RNA toxins coevolving with CRISPR-Cas, which underlie the selfish feature of CRISPR-Cas systems, and illuminated a specific and robust strategy combining CRISPR antimicrobials and TA antimicrobials, which can be expected to enrich our arsenal to combat the increasingly threatening antibiotic resistance.

## Methods

### Bacterial strains and growth conditions

Strains used in this study are listed in Supplementary Data. 3. *E. coli* DH5α was used for plasmid construction and *E. coli* s17-1 λ pir was used as the donor strain for conjugation. *A. baumannii* clinical isolates, numbered from Aba1 to Aba18, with their drug susceptibility and genomic information, were provided by Jie Feng's lab at Institute of Microbiology, Chinese Academy of Sciences (see Supplementary Data. 1). These clinical isolates were used to test the versatility of WCHA45 CreTA and *A. baumannii* AYE CRISPR-Cas (Supplementary Fig. 10), and Aba14 was used for the ATTACK assay (Fig. 6). *A. baumannii* ATCC19606, *A. baumannii* ATCC17978, WT *A. baumannii* AYE (ΔpyrF) and its *cas* mutants (Δcas1, Δcas3, and Δcsy) were from a previous study[35]. The AYE mutants TA-in, TA-out, Δcsy4, and Δaac3 were constructed in this study, as previously described[35] (also see below).

All bacterial strains were grown at 37 °C in lysogeny broth (LB) media (10 g/L tryptone, 5 g/L yeast extract, 10 g/L NaCl), with agar (15 g/L) for solid plates and with 200 rpm shaking for liquid cultures. When required, media were supplemented with uracil (50 µg/ml),

5-fluoroorotic acid (5-FOA; 50 µg/ml), kanamycin (50 µg/ml), potassium tellurite (30 µg/ml), apramycin sulfate (50 µg/ml), gentamicin (8 µg/ml, unless specified), imipenem (32 µg/ml), or meropenem (4 µg/ml).

### Plasmid construction

Plasmids and oligonucleotides used in this study are listed in Supplementary Data. 3, all oligonucleotides were produced from Tsingke Biological Technology, Beijing, China. The conjugative plasmid pMo130TFR and its derivate pMo130TFRI (has a *lacI* gene) were from a previous study[35], and used for gene expression in this study. The *creTA* genes of *A. baumannii* WCHA45, LoGeW2-3, and ANC3789 were amplified using Phanta Max Super-Fidelity DNA Polymerase (Vazyme Biotech Co.,Ltd). When required, nucleotide mutations or deletions were achieved via overlapping PCR (Figs. 2, 3d, 3e and Supplementary Figs. 1, 2). For CreT characterization, a synthetic promoter $P_{tac}$ was used to control its expression (Fig. 4 and Supplementary Fig. 4). For IPTG-inducible expression, a modified $P_{tac}$ combined with a *lacO* operator was used, and pMo130TFRI (has a *lacI* gene) was used as the expression vector (Fig. 4e and Supplementary Fig. 6). These constructs were digested by BamHI and PstI (New England Biolabs, MA, USA), and then ligated to predigested plasmids using the T4 DNA ligase (New England Biolabs, MA, USA).

For the fluorescence assay (Fig. 3b, c and Supplementary Figs. 4b, 5), *gfp* and the *creT* elements were separately amplified, and assembled into predigested plasmids through a three-piece Gibson assembly strategy using Trelief® Seamless Cloning Kit (Tsingke Biological Technology, Beijing, China).

To construct pAAC3 (Fig. 5), a mini-CRISPR consisting of the leader sequence of AYE CRISPR, two repeats, and an *aac3*-targeting spacer was assembled via overlapping PCR. The PCR products were digested by BamHI and PstI (New England Biolabs, MA, USA), and ligated to predigested pMo130TFR. To construct pAAC3-TA (Fig. 5a), The WCHA45 *creTA* genes were amplified, digested by ApaI and then ligated to predigested pAAC3. Note that, the two-piece Gibson assembly strategy could also be adopted, taking advantage of the homologous arms designed on related primers.

To construct the AYE mutants TA-in and TA-out (Fig. 5b), wherein the WCHA45 *creTA* was inserted between the *cas3* and *csy1* genes or at the original genomic site of the deleted *pyrF* gene, the *creTA* genes and the 5′- and 3′-regions (each 1 kb) flanking the target site were separately amplified, and then inserted into predigested (by BamHI and PstI) pMo130TF (a suicidal vector[35]) using a four-piece Gibson assembly strategy.

To knock out *csy4* (Fig. 2b and Supplementary Fig. 2) or *aac3* (Supplementary Fig. 7) from AYE, the 5′- and 3′-flanking sequences (each 1 kb) were amplified and assembled into pMo130TF[35] using the three-piece Gibson assembly strategy. To complement a plasmid copy of *aac3*, its native promoter and coding sequence were amplified and inserted into predigested (by BamHI and PstI) pMo130TFR or pCasAb-Apr[36].

To construct pCas (Supplementary Fig. 10), the fragment containing the *rep* gene, *oriT*, ori-pBR322, and the tellurite-resistance gene was amplified from pMo130TFR, and the fragments containing the promoter of *cas1* and the *cas3-csy* gene cluster were amplified from the AYE genome. These fragments were assembled using a three-piece Gibson assembly strategy to generate pCAS. Then, to construct pCasT or pCasTA (Supplementary Fig. 10), the sequence of WCHA45 *creT* or *creTA* (ΨR2 replaced by AYE repeat) was amplified and inserted into the ApaI site of pCAS via a two-piece Gibson assembly strategy. To construct pATTACK (Fig. 6), an *oxa23*-targeting mini-CRISPR controlled by the J23114 promoter and *lacO* operator was engineered via overlapping PCR. This fragment and the *lacI* gene were assembled into pCas or pCasTA (predigested by BamHI and PstI) using the three-piece Gibson assembly strategy.

## Plasmid transformation

Bacterial cells were transformed using the electro-transformation method, unless specified. The electrically competent cells of *A. baumannii* were prepared by inoculating a colony into liquid medium and the culture was grown at 37 °C overnight with 200 rpm agitation. The overnight culture was sub-inoculated into fresh LB medium with a ratio of 1:50, and then cultured until OD600 reaching to 0.4–0.6. Cultures were placed on ice and the log phase cells were washed three times with 10% glycerol to make them electrocompetent. Then, 300 ng plasmids were transformed into 100 µl of competent cells using the bacteria program on the MicroPulser Electroporator (Bio-Rad, CA, USA). After 1 h of recovery in 500 µl of LB, the culture was tenfold serially diluted by LB and streaked on plates containing potassium tellurite and/or antibiotics. Transformation efficiency was calculated by counting the colonies on plates and multiplying the value by the dilution ratio. Every assay has been conducted with three biological replicates. The average and standard deviation of transformation efficiency (colony forming unit per µg plasmid DNA, CFU/µg) were calculated based on log-transformed data.

For the conjugation assay (Fig. 6), plasmids were first introduced into the *E. coli* S17-1 λ pir cells. Stationary cells of *E. coli* S17-1 and *A. baumannii* were mixed (10:1) and washed twice with water to remove potassium tellurite. The mixture was spotted onto a 0.22 µm sterile filter membrane in the center of solid LB plates. After incubating at 37 °C for 8 h, the mixture was resuspended, diluted and coated on different selective LB plates. When needed, 0.5 mM IPTG was added to induce CRISPR expression. Transformation efficiency was calculated by counting the colonies on plates and multiplying the value by the dilution ratio. Every assay has been conducted with six biological replicates.

## Strain construction

A *pyrF*-based Efficient Genetic Manipulation Platform[35] was used for gene knock-in or knock-out in *A. baumannii* AYE. Plasmids used for gene knock-in or knock-out were constructed as described above. AYE cells were transformed using the conjugation strategy (also described above), and plated on LB medium containing potassium tellurite to obtain single-crossover colonies. After validation by colony PCR, at least two colonies were inoculated into liquid medium containing uracil and 5-FOA. Then, the stationary phase culture was streaked on LB plates containing uracil and 5-FOA, and the resulting colonies were randomly selected for replica plating. Colonies that grew on LB plates lacking potassium tellurite, but not on plates containing potassium tellurite, were regarded as double-crossover candidates. PCR amplification and DNA sequencing were finally performed to screen and validate the genotype, respectively.

## Primer extension analysis

The 5'-FAM (6-carboxyfluorescein)-labeled *gfp*-specific primer (Supplementary Data. 3) was ordered from Sangon Biotech (Shanghai) Co., Ltd.. 5 µg of the total RNA was firstly digested with RQ1 DNase (Promega, WI, USA), and then reverse transcribed into complementary DNA (cDNA) using 30 enzyme units(U) of the Moloney Murine Leukemia Virus reverse transcriptase (MMLV-RT) (Promega, WI, USA) and 2.5 µg of the labeled primer. The extension products were analyzed using the ABI3730xl DNA Analyzer (Thermo Fisher Scientific, MA,USA), and the results were viewed using Peak Scanner Software v1.0.

## IS insertion analysis

The gentamicin resistant colonies obtained from the pAAC3 and pAAC3-TA transformation assays were randomly selected for colony PCR. Primers *csy*-F and *csy*-R (Supplementary Data. 3) were used for PCR amplification, and the PCR products were subjected to agarose gel electrophoresis (0.8%) and Sanger DNA sequencing. The sequencing results were viewed using the SnapGene software (version 3.2.1) (Supplementary Fig. 8).

## RNA extraction and Northern blotting

The log phase cells of *A. baumannii* AYE were collected for RNA extraction. The total RNA was prepared using the TRIzol reagent (Thermo Fisher Scientific, MA, USA), purified using the phenol: chloroform method, precipitated with equal-volume ethanol, and stored at −80 °C.

For Northern blotting analysis, the RNA samples were dissolved in RNA-free water and quantified using Nanodrop 2000 (Thermo Fisher Scientific, MA, USA). For each sample, 10 µg RNA were mixed with an equal volume of 2 × RNA loading dye (New England Biolabs, MA, USA) and denatured at 65 °C for 10 min.

The denatured RNA samples were loaded onto an 8% poly-acrylamide gel (7.6 M urea) and electrophoresed in 1× TBE buffer at 200 V, together with biotin-labeled single-stranded DNA (serving as a custom size marker) and the Century-Plus RNA ladder (Thermo Fisher Scientific, MA, USA). After electrophoresis, the RNA ladder lane was excised, stained with Ultra GelRed (Vazyme Biotech, Nanjing, China), and then imaged. RNA samples in the remaining gel were transferred to a Biodyne B nylon membrane (Pall, NY, USA) using the Mini-Protean Tetra system (Bio-Rad, CA, USA). After cross-linking using UV light, the membrane was hybridized with biotin-labeled DNA probes. The biotin signals were detected with the Chemiluminescent Nucleic Acid Detection Module Kit (Thermo Fisher Scientific, MA, USA) according to the manufacturer's protocol, and imaged using the Tanon 5200 Multi chemiluminescent imaging system (Tanon Science & Technology, Shanghai, China).

## Small RNA sequencing

Total RNA was extracted from *A. baumannii* AYE cells expressing the WCHA45 *creTA* (ΨR2-replaced) as described above. A total of 50 µg of RNA was treated with T4 polynucleotide kinase (New England Biolabs, MA, USA) according to the manufacturer's protocol at 37 °C. And then the kinase was inactivated by incubating at 65 °C for 20 min. The treated RNA was purified using the phenol-chloroform method, precipitated with an equal volume of isopropanol and 0.1 volume of 3 M sodium acetate. The RNA sample was re-dissolved in RNA-free water for sequencing. A small RNA library was constructed for the RNA molecules ranging from 30 to 300 nt with the NEXTFLEX Small RNA-Seq Kit (Bioo Scientific, TX, USA), and then subjected to Illumina HiSeq sequencing (paired-end, 150 bp reads). The raw data reads were processed to remove adapters and mapped to the *creA* sequence using previously reported Perl scripts[19].

## Fluorescence measurement

For each experimental setting, three individual colonies were randomly selected for this assay. *A. baumannii* AYE cells expressing *gfp* were cultured to the late exponential phase, and their OD600 and fluorescence were simultaneously determined using the Synergy H4 Hybrid multimode microplate reader (BioTeck, VT, USA). The fluorescence/OD600 ratio was calculated for each of the three individual biological samples.

## qPCR

50 µl of the exponential AYE culture were subjected to fluorescence measurement as described above, and the remaining culture to total RNA extraction. A total of 10 µg RNA was firstly treated with RQ1 DNase (Promega, WI, USA) to remove the genomic DNA according to the manufacturer's instruction. Then, the DNA removed RNA samples were purified using the phenol-chloroform method, and reverse transcribed into complementary DNA (cDNA) using the Moloney Murine Leukemia Virus reverse transcriptase (MMLV-RT) (Promega, WI, USA) and Random Hexamer Primer (Thermo Fisher Scientific, MA,

USA). qPCR assay was performed using the KAPA SYBR® FAST qPCR Kit (Kapa Biosystems, MA, USA) and run on an Applied Biosystems ViiA™ 7 Real-Time PCR System. The *rho* gene was used as a loading control. For each experimental setting, three biological replicates (from individual colonies) were included, and each biological replicate was examined in triplicate. The primers used for qPCR are listed in Supplementary Data. 3.

### Dilution platting assay

*A. baumannii* AYE cells containing an inducible *creT* were cultured overnight in LB medium (containing 30 μg/ml potassium tellurite). The overnight culture was sub-inoculated into fresh medium and cultured to stationary phase (~6 h). For the assay in Fig. 4e, 0.5 mM IPTG or equal volume of LB medium (as a control) was added into the culture. After 0, 0.5, 1, 2, 4, or 6 h, AYE cells were sampled, and then washed and serially diluted by LB medium, after which 2 μl of each dilution were spotted on solid LB plates with or without 0.5 mM IPTG. For the assay in Supplementary Fig. 6, the stationary culture was directly subjected to serial dilution and then plated on medium containing varying amounts of IPTG (0, 0.1, 0.2, 0.4, 0.8, and 1.6 mM).

### Codon usage analysis

The *A. baumannii* AYE (NCBI ID: NC_010410.1) coding genes were downloaded from the NCBI (https://ftp.ncbi.nlm.nih.gov/genomes/all/GCA/000/069/245/GCA_000069245.1_ASM6924v1/), and the codon usage was analyzed by using the cusp program on EMBOSS explorer (http://emboss.toulouse.inra.fr/cgi-bin/emboss/cusp).

### Search for *creTA* homologs and analogs

We initially identified a potential *creTA* element in the *cas3-csy1* intergenic region (IGR) in *Acinetobacter* sp. ANC3789. Using this IGR sequence as a query, we searched the NCBI nucleotide database with the Blastn program and found its homologs only in *Acinetobacter* sp. WCHA45 and LoGeW2-3. To find more homologs, we searched the NCBI protein database using WCHA45 Cas3 and Csy1 proteins, and retrieved the IGR sequences intervening their most related homologous proteins. By manually examining these IGR sequences, we further discovered 19 more *creTA* modules.

### Bioinformatic analysis

Sequence alignments were constructed and viewed using the GeneDoc software (version 2.6.002). RNA secondary structure was predicted using the RNAfold webserver. Putative RNA interactions were predicted using the IntaRNA webserver (Freiburg RNA Tools). Promoter elements were predicted using the BPROM program (Softberry tool).

### Data analysis and image visualization

Microsoft Excel was used to analyze the data, and GraphPad Prism (version 7.00) was used to generate the plots. The graphs were then modified in Adobe Illustrator (version 24.0) to construct the final figures.

### Statistics & reproducibility

All experiments in the present manuscript were repeated at least three times independently with similar results, including those in Figs. 2b, c, 3b–e, 4a, 4c–e, 5a, b, 6c, Supplementary Figs 1a, b, 2, 3, 4a, b, 5b, 6b, 8a, b, and 9b.

### Reporting summary

Further information on research design is available in the Nature Portfolio Reporting Summary linked to this article.

### Data availability

All relevant data are included in the paper and/or its supplementary information files. Source data are provided as a Source Data file. The raw data for the RNA-seq experiments in Fig. 2 were deposited to the National Center for Biotechnology Information (NCBI) with the Bio-Project accession number PRJNA950041. All strains and plasmids are available from the corresponding author upon request; requests will be answered within 2 weeks. Source data are provided with this paper.

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

## Acknowledgements

This work was supported by the National Natural Science Foundation of China [32150020 (M.L.), 32022003 (M.L.), 32270092 (R.W.), 32200057 (F.C.), 31970544 (M.L.)], Science & Technology Fundamental Resources Investigation Program [2022FY101100] (J.F. and H.Z.), the Strategic Priority Research Program of the Chinese Academy of Sciences (Precision seed design and breeding) [XDA24000000] (M.L.), the Youth Innovation Promotion Association of CAS [2020090] (M.L.), and the National Postdoctoral Program for Innovative Talents of China [BX20220331] (F.C.).

## Author contributions

M.L. and R.W. conceptualized this study. M.L., R.W., and X.S. designed the experiments, with valuable suggestions from H.Z., F.C., and Q.X.. R.W., X.S., H.Z., C.L., A.W., L.W., and Y.Z. constructed creTA plasmids and performed bacteria transformation assays. R.W., X.S., H.Z., and F.C. conducted the MDR pathogen-killing assays. C.L., F.C., Q.X., and R.W. performed RNA-seq, Northern blotting, primer extension and gene knock-in/knock-out assays. X.S. conducted the fluorescence measurement and qPCR assays. R.W. and X.S. preformed the dilution plating assay. J.F. provided clinical isolates and the information of their genome and drug susceptibility. M.L., R.W., and X.S. analyzed the RNA-seq data and performed the bioinformatic analyses. Formal analysis of results was done by R.W., X.S, H.Z., and F.C.. N.W. contributed to discussions and provided feedback throughout the project. M.L. analyzed the data and wrote the paper, which was edited and approved by all authors.

## Competing interests

M.L., R.W., X.S., and F.C. have filed a patent on the ATTACK antimicrobial with the institution as the applicant (application number: 202210947272.7; pending). N.W. is a cofounder of CreatiPhage Biotechnology. The other authors declare no competing interests.
