## [Peer Review File · Nature Communications]

Reviewers' Comments:

Reviewer #1:

Remarks to the Author:

Wang and colleagues characterize a novel CRISPR-regulated toxin-antitoxin (CreTA) system in *Acinetobacter*. This is the first bacterial CreTA characterized, and the manuscript's findings are important with various implications for this alone. Out of three bacteria having putative creTA systems, they show that, for one, the system can be transferred with functionality to another strain. They map the boundaries of the CreA antitoxin. Using a GFP reporter, they convincingly show that a "spacer", which acts as the antitoxin, binds upstream of the toxin and negatively regulates its transcription. They further show elegantly that the homology of the CreA with the putative target is required for suppressing it. Complementation experiments of the homology undoubtedly show the interactions. The authors further show that the toxin is not a protein but rather an RNA that most likely binds to the 23S rRNA – interfering with ribosomal function. Moreover, they show that the toxicity is lytic rather than static, unlike the CreTA from archaea. The characterization of the CreTA system identified here is complete and technically sound. Lastly, the authors propose to use the CreTA as an antimicrobial in combination with the CRISPR system. Based on their finding, they propose that a CRISPR-based antimicrobial would be more efficient than CRISPR alone. They suggest that bacteria escaping the targeting due to CRISPR inactivity will not survive because they will be targeted by the toxin (in the absence of creA which is activated by the CRISPR system). Overall, this approach seems appealing. Some experiments and clarifications are needed to conclude whether and how much it improves CRISPR-antimicrobials' efficiency.

The manuscript is well written, and the general findings convincingly support the conclusions. As indicated above, the thorough and complete characterization of the bacterial CreTA system itself merits publication. However, the ATTACK strategy needs some more clarifications before publication.

Major concerns:

1. Line 376 – the lytic phages not only target those bacteria that evolved resistance but also those that were not treated in the first place with the sensitizing CRISPR-anti-microbials. This is not present in the current strategy. The current strategy shows efficient eradication of antibiotic-resistant bacteria, but only upon antibiotic selection of the conjugant\transformed bacteria. How do the authors propose to target bacteria that did not conjugate? Those that the CRISPR antimicrobials did not reach? In addition, the lytic phage strategy targets plasmids and does NOT generate counter-selective pressure. In the first place, such a strategy does not require active maintenance of the CRISPR-antimicrobials. The authors might want to clarify this and present each strategy's shortcomings and advantages.

2. What is the benefit of the ATTACK strategy in the absence of counter-selection against the CRISPR-antimicrobials? The authors may wish to design a similar experiment to that presented in Figure 6, but instead of targeting the genome, which leads to counter-selection, they should target a plasmid whose elimination does not reduce the survival of the bacteria. They then should assess the effect of the TA module. Is it also useful under such conditions?

3. Line 307 – the authors look at IS insertions and show that they are not present in the presence of the Cre-TA module. Nevertheless, they should sequence other elements in the surviving bacteria to have a complete picture. The obvious ways to escape the toxicity of the CRISPR-antimicrobials are to mutate the protospacers targeted by the CRISPR. The authors should pick 30-50 bacteria from each type of experiment (preferably from independent experiments – to avoid siblings) and sequence their targeted protospacer. It is expected that in the absence of the creTA module – there will be significantly fewer escapees with such mutations because inactivation of the CRISPR is possible.

On the other hand, in the group with the creTA module - much more mutations in the protospacer should be detected, because this is a major way to avoid the antimicrobials. Some of the data shows that the TA module increases selection by 20-fold (line 318 – from 471 fold to 10100 fold). This means that ~95% of the CRISPR is lost when the TA module is absent. Such numbers should also be reflected in the sequencing of the targeted protospacer.

Minor comments:

4. Line 48 – not only fragments of viruses. Also plasmids, genome, and unclassified DNA is inserted into spacers.
5. Line 123 – Dalgarno (not Dalgarna).
6. Line 227 – the word “decide” should be replaced. Perhaps “dictate”?
7. Line 227 – perhaps indicate that the apparent contradiction between the requirement for Shine-Dalgarno sequence, and the apparent contradiction that no protein is required for activity is to be explained in the next paragraph. Otherwise, the reader is confused at this point.
8. Line 309 – consider rewording the sentence “It was indicated that...”.
9. Line 343 – “filter” (not filer).

Reviewer #2:

Remarks to the Author:

In this manuscript, Wang et al describe the characterization of the first bacterial CRISPR-regulated toxin-antitoxin (TA) system of type VIII, CreTA, associated with type I-F CRISPR-Cas in *Acinetobacter* and its application for a new antimicrobial strategy.

The authors emphasize an important point on the recently discovered two-RNA TA system and its functional link with CRISPR-Cas with CreT bactericidal RNA toxin and CreA antitoxin guiding CRISPR effector for CreT repression. They provide proof-of-concept for an efficient antimicrobial strategy against multi-drug resistant pathogens like *Acinetobacter baumannii* combining CRISPR-Cas targeting of antibiotic resistance genes and CreTA triggering the cell death if CRISPR-Cas fails.

Overall, the paper is well written and the experiments are well performed.

The paper represents an important advancement for the characterization of TA systems and their link with CRISPR-Cas opening interesting perspectives for future studies.

The study represents a large amount of work for the detailed analysis of the first bacterial CreTA system. On my opinion, it would be interesting to complete if possible this study with evaluation of the mechanisms mediating the CreT action to strengthen the hypotheses discussed in the paper.

I have some concerns that should be addressed prior to publication.

Major comments:

1. Figure 1b depicts the creTA region. Please include the promoter elements for creA and creT divergent promoters as well as terminators on this figure. Only TSS-creA is indicated, how it was defined? What about TSS-creT?
2. The authors analysed the expression of creA (wild-type and modified sequences) by Northern blotting and small RNA-seq (Figure 2 and Suppl Fig 1). I wondered whether the creT expression could be monitored?
3. Lines 130-149: before describing the repeat replacement, please explain and show how the toxicity of creT from this new TA module has been demonstrated as well as the antitoxin action of wt creA. CreT toxicity is shown only later in the manuscript (Fig 4 and Supplm Fig 8 for different *Acinetobacter* isolates). To demonstrate the functional link with CRISPR-Cas machinery as for previously described creTA, the transformation efficiency in delta cas background is presented in Suppl Fig 2 and mentioned only at the end of the paragraph. I would suggest indicating these results from the beginning when describing the identification of a new CreTA associated with CRISPR-Cas.
4. An important remaining question is the nature of CreT. The authors tried to assess this point in the manuscript by mutational analysis but it remains still unclear if it acts as an RNA or as a small peptide translated from mini-ORF (“its protein product (if it does exist)” conclusion line 237-238)? I wondered whether the product/active translation could be detected in vitro (for example in vitro transcription/translation system) and in *Acinetobacter* (global approaches like riboprofiling or small peptides proteomics)? The authors checked the functionality of SD region associated with mini-ORF and discussed the results (lines 212-218) presented on Suppl Fig 3. On my opinion it is important to mention this earlier in the text when describing first gfp fusions (paragraph starting with line 164).

5. The use of *gfp* reporter gene fusion constructs should be better explained in the following paragraph "CreA transcriptionally repress its cognate *creT*". It is not clear if transcriptional or translational fusions are used in these experiments. To check the activity of promoter and transcriptional effect of *creA* on it, the authors should construct a transcriptional fusion carrying only promoter regulatory elements from the gene of interest and keeping canonical translational initiation signals for *gfp*. When using the constructs carrying both the promoter elements and SD to initiate translation both transcriptional and post-transcriptional/translational effects could be seen. The constructs presented should be combined with the data on Suppl Fig 3 with *ptac* promoter and mini-ORF SD.

On my opinion, the paragraph starting from the line 174 should be modified. Fluorescence measurement and *gfp* RNA abundance analysis could not directly respond to the question about the translational effects. Please clarify this part including the Supplm Fig 3 constructs and additional combinations of regulatory elements.

6. Line 183: the title should be modified: "does not have a canonical seed"

7. Lines 267-268: The authors concluded that the toxicity of *CreT* seems to correlate with the efficiency of translational initiation, but direct evidence for this correlation is still missing. On my opinion, it is important to try to detect the active translation or protein product to conclude or modulate the statement.

8. The paragraph "CreT is bactericidal small RNA likely targeting ribosomal RNAs" should be modified. On my opinion, the part on *CreT* toxicity (lines 281-287) should be moved to the first part of the manuscript for the description of TA module and the title of the paragraph should be modified accordingly. The authors claim that *CreT* interacts with 16S RNA and possibly with 23S rRNA (lines 278-279). I wondered whether the authors tried to identify the direct targets of *CreT*? To strengthen this conclusion the targets need to be identified experimentally. I would suggest at least including some elements on the *CreT* mechanism of action in the discussion section.

9. The authors claim that *CreTA* improved the efficiency of CRISPR antimicrobials (line 351-352), the difference is not very high (97.84% to 98.96%), is it statistically significant? Please specify.

10. For the application of this antimicrobial strategy it is important to assess the long-term effects through several generations during bacterial growth. Has the survival over the more long-term been tested?

Minor Issues:

- It could be helpful including page number to the manuscript.
- The delay in providing supplementary data on submission was not helpful for the paper reviewing.
- "x-log reduction" to avoid throughout the manuscript (Line 139, 144, 145, 192, 196, 200 ...)
- Line 60 "Clostridium difficile" is a previously used name, now "Clostridioides difficile" is currently used, please check
- Line 123 Please modify "Shine-Dalgarna" to "Shine-Dalgarno"
- Line 153 and 232 "wild-type"
- Line 296 "(bearing an *aac3*-targeting CRISPR)" please specify "mini-CRISPR arrays with spacer targeting the *aac3*"/specific crRNA ? The same for line 338 "oxa23-targeting CRISPR".
- Line 298 Please explain why "potassium tellurite" is used for selecting for all transformants.
- Fig. 1 Please include the position of promoter region/TSS/terminator for *creA* and *creT*.
- Line 621 "10 mg RNA" please check, I think it should be "10 microgram".

Reviewer #3:

Remarks to the Author:

In the manuscript entitled "ATTACK: Associate Toxin-Antitoxin with CRISPR-Cas to Kill Multidrug-Resistant Pathogens", Wang et al reported and characterized the first bacterial *CreTA* system, established its working mechanism, and applied the system to boost the CRISPR-mediated antimicrobials in *A. baumannii*. The work is highly innovative with both new fundamental principles and substantial translational values. I have a few points for the authors.

1. The authors only characterised the system and applied the system in *A. baumannii*, rather than several common MDR pathogens, and it is not straightforward to extend the system to other

pathogens. Hence, it would be appropriate to change the title to "...kill Multidrug-Resistant *Acinetobacter Baumannii*".

2. A large number of CRISPR-Cas systems in bacteria are degenerated or are inhibited by Acrs. My understanding is that the ATTACK system works only in those pathogens containing an active CRISPR-Cas system, not those with a degenerated CRISPR-Cas system or those containing Acrs. Authors should clarify this point.

3. The authors claim that ATTACK is applied to eliminate MDR pathogens, what are the MDR profiles of the strains they used in this study.

4. What is the potential of the application of ATTACK system in vivo? Authors may discuss this point in the Discussion.

5. A schematic illustration of the working principle of the test should be provided in Fig 2

Minor points:

There are grammar and sentence construction problems in the text, I am listing some of them:

1) L 30 and L354-355, "kill pathogens and cure resistance genes" can not be written in short as "kill or cure MDR pathogens"

2)L 44, appearance should be "emergence"

3)L 58, cure them, "them" mean pathogens or infection? Infections can be cured, but not pathogens.

4) L124, runs should be "is oriented"

5) L135, make should be "enable"

6) L227, decide should be "dictate"

7) 253, disappeared should be "was lost"

8)L 277, kept should be "retained"

9)L312, partly should be "partially"

10) L338, put should be "place"

Point-by-point Response to Referees

Response to Reviewer #1:

Wang and colleagues characterize a novel CRISPR-regulated toxin-antitoxin (CreTA) system in *Acinetobacter*. This is the first bacterial CreTA characterized, and the manuscript's findings are important with various implications for this alone. Out of three bacteria having putative creTA systems, they show that, for one, the system can be transferred with functionality to another strain. They map the boundaries of the CreA antitoxin. Using a GFP reporter, they convincingly show that a “spacer”, which acts as the antitoxin, binds upstream of the toxin and negatively regulates its transcription. They further show elegantly that the homology of the CreA with the putative target is required for suppressing it. Complementation experiments of the homology undoubtedly show the interactions. The authors further show that the toxin is not a protein but rather an RNA that most likely binds to the 23S rRNA – interfering with ribosomal function. Moreover, they show that the toxicity is lytic rather than static, unlike the CreTA from archaea. The characterization of the CreTA system identified here is complete and technically sound.

Lastly, the authors propose to use the CreTA as an antimicrobial in combination with the CRISPR system. Based on their finding, they propose that a CRISPR-based antimicrobial would be more efficient than CRISPR alone. They suggest that bacteria escaping the targeting due to CRISPR inactivity will not survive because they will be targeted by the toxin (in the absence of creA which is activated by the CRISPR system). Overall, this approach seems appealing. Some experiments and clarifications are needed to conclude whether and how much it improves CRISPR-antimicrobials' efficiency.

The manuscript is well written, and the general findings convincingly support the conclusions. As indicated above, the thorough and complete characterization of the bacterial CreTA system itself merits publication. However, the ATTACK strategy needs some more clarifications before publication.

Response: Thank you for the positive evaluations and helpful suggestions.

Major concerns:

1. Line 376 – the lytic phages not only target those bacteria that evolved resistance but also those that were not treated in the first place with the sensitizing CRISPR-anti-microbials. This is not present in the current strategy. The current strategy shows efficient eradication of antibiotic-resistant bacteria, but only upon antibiotic selection of the conjugant/transformed bacteria. How do the authors propose to target bacteria that did not conjugate? Those that the CRISPR antimicrobials did not reach? In addition, the lytic phage strategy targets plasmids and does NOT generate counter-selective pressure. In the first place, such a strategy does not require active maintenance of the CRISPR-antimicrobials. The authors might want to clarify this and present each strategy's shortcomings and advantages.

Response: Thank you for this suggestion. We agree that the ATTACK strategy could not kill pathogens that do not receive the CRISPR-Cas and CreTA elements. In fact, delivery efficiency and targeting efficiency are two main bottlenecks of CRISPR antimicrobials. Our ATTACK strategy aims to elevate the targeting efficiency by improving the genetic stability (and possibly also expression level and activity) of CRISPR-Cas components in the cell population.

Compared to ATTACK, the lytic phage strategy does have its advantage in killing pathogens that CRISPR antimicrobials do not reach, which was discussed in Line 418. Notably, unlike the well-characterized phages infecting model *E. coli* strains, engineered phages have not been developed for the clinically important *A. baumannii* isolates, so here we choose the conjugation-based delivering method to demo the potential of ATTACK strategy. We believe with engineered *A. baumannii* phages, future combination of the two strategies will give rise to more effective antimicrobials (discussed in Line 420).

About the counter-selective pressure, we think it mainly depends on the location of target gene (on chromosome or plasmid), not on the different strategies.

2. What is the benefit of the ATTACK strategy in the absence of counter-selection against the CRISPR-antimicrobials? The authors may wish to design a similar experiment to that presented in Figure 6, but instead of targeting the genome, which leads to counter-selection, they should target a plasmid whose elimination does not reduce the survival of the bacteria. They then should assess the effect of the TA module. Is it also useful under such conditions?

Response: We selected to target the chromosomal AR genes because most AR genes in *A. baumannii* isolates locate on their chromosome (e.g. AYE has an 86-kb resistant island).

Supplementary Fig. 9

As suggested, we tested the effects of CreTA in curing a plasmid-carrying AR gene. Because we could not find a natural plasmid carrying AR genes in the *A. baumannii* strains/isolates in our hands, we selected to target an engineered plasmid carrying the *aac3* gene. From the data (Supplementary Fig. 9), we can see that CreTA increased the effect (reducing antibiotic-resistant colonies) of CRISPR antimicrobials on medium containing medium-level gentamycin (4 $\mu\text{g}/\text{ml}$), but interestingly not on medium containing high-level gentamycin (8 $\mu\text{g}/\text{ml}$). Apparently, the combination of CRISPR and CreTA achieved more complete AR curing. We inferred that CreTA not only protects CRISPR-Cas from being inactivated (e.g., by IS elements), but also ensures its high-level expression and targeting activity (CRISPR-Cas activity may be compromised by genetic or non-genetic factors in the absence of CreTA, which lead to incomplete AR curing and survival on lower level of selection).

3. Line 307 – the authors look at IS insertions and show that they are not present in the presence of the Cre-TA module. Nevertheless, they should sequence other elements in the surviving bacteria to have a complete picture. The obvious ways to escape the toxicity of the CRISPR-antimicrobials are to mutate the protospacers targeted by the CRISPR. The authors should pick 30-50 bacteria from

each type of experiment (preferably from independent experiments – to avoid siblings) and sequence their targeted protospacer. It is expected that in the absence of the creTA module – there will be significantly fewer escapees with such mutations because inactivation of the CRISPR is possible. On the other hand, in the group with the creTA module - much more mutations in the protospacer should be detected, because this is a major way to avoid the antimicrobials. Some of the data shows that the TA module increases selection by 20-fold (line 318 – from 471 fold to 10100 fold). This means that ~95% of the CRISPR is lost when the TA module is absent. Such numbers should also be reflected in the sequencing of the targeted protospacer.

Response: We agree that mutation of the target protospacer is the major way to avoid CRISPR antimicrobials, which was supported by our data (Figure 5) that >99% of the survivors were re-sensitized and could not grow on selective Gen⁺ medium, no matter the *creTA* module was present or not. By examining the colonies grown on non-selective medium (most were AR- survivors), we verified that the *aac3* gene (probably the entire resistant island) was missing in the most majority (see the Figure below).

Detection of *aac3* by colony PCR.

In fact, in Supplementary Fig. 8, we specifically investigated the AR⁺ survivors (grown on Gen⁺ medium), most of which should have escaped CRISPR antimicrobials in ways other than target mutation.

CRISPR-Cas activity could be abrogated by various genetic variations within *cas* genes or their regulating genes/elements, like spontaneous point mutations, MGE transpositions (including IS elements), etc., and can also be compromised by some non-genetic factors to survive the conflict between CRISPR-Cas and antibiotic resistant genes (just as indicated by the plasmid curing assay; see above and Supplementary Fig. 9). It was also recently reported that resistance frequently arise from mutations disrupting the CRISPR-Cas antimicrobial or its regulatory elements (lower Cas expression could also contribute to resistance) (Scientific Reports, (2021) 11:17267). These genetic and non-genetic variations could hardly be fully analyzed in this study. Disruption by IS insertion represents only one of these resistance mechanisms, which we could readily analyze to exemplify the protective effect of CreTA on CRISPR antimicrobial.

Minor comments:

4. Line 48 – not only fragments of viruses. Also plasmids, genome, and unclassified DNA is inserted into spacers.

Response: Corrected correspondingly.

5. Line 123 – Dalgarno (not Dalgarna).

Response: Corrected correspondingly.

6. Line 227 – the word “decide” should be replaced. Perhaps “dictate”?

Response: Corrected correspondingly.

7. Line 227 – perhaps indicate that the apparent contradiction between the requirement for Shine-Dalgarno sequence, and the apparent contradiction that no protein is required for activity is to be explained in the next paragraph. Otherwise, the reader is confused at this point.

Response: This paragraph was revised to explain the contradiction (Line 241).

8. Line 309 – consider rewording the sentence “It was indicated that...”.

Response: Reworded (Line 314).

9. Line 343 – “filter” (not filer).

Response: Corrected correspondingly.

Response to Reviewer #2:

In this manuscript, Wang et al describe the characterization of the first bacterial CRISPR-regulated toxin-antitoxin (TA) system of type VIII, CreTA, associated with type I-F CRISPR-Cas in *Acinetobacter* and its application for a new antimicrobial strategy.

The authors emphasize an important point on the recently discovered two-RNA TA system and its functional link with CRISPR-Cas with CreT bactericidal RNA toxin and CreA antitoxin guiding CRISPR effector for CreT repression. They provide proof-of-concept for an efficient antimicrobial strategy against multi-drug resistant pathogens like *Acinetobacter baumannii* combining CRISPR-Cas targeting of antibiotic resistance genes and CreTA triggering the cell death if CRISPR-Cas fails.

Overall, the paper is well written and the experiments are well performed.

The paper represents an important advancement for the characterization of TA systems and their link with CRISPR-Cas opening interesting perspectives for future studies.

The study represents a large amount of work for the detailed analysis of the first bacterial CreTA system. On my opinion, it would be interesting to complete if possible this study with evaluation of the mechanisms mediating the CreT action to strengthen the hypotheses discussed in the paper.

Response: Much appreciation for your positive evaluations and helpful suggestions.

I have some concerns that should be addressed prior to publication.

Major comments:

1. Figure 1b depicts the *creTA* region. Please include the promoter elements for *creA* and *creT* divergent promoters as well as terminators on this figure. Only TSS-*creA* is indicated, how it was defined? What about TSS-*creT*?

Response: Figure 1 was revised as suggested. TSS-*creA* was identified during RNA-seq analysis (see Fig. 2c). Because the reads mapped to *creA* were very low in abundance (~20 per million reads), only its processing site and TSS could be readily detected (terminator could not be determined).

For the same reason, TSS of *creT* was not convincingly detected by RNA-seq (its transcription was suppressed by CreA and its RNA products may be further processed by unknown nucleases). Instead, we further performed primer extension to identify the TSS of *creT* (see the new data in Supplementary Fig. 3 and related statements in line 165-174).

Supplementary Fig. 3

2. The authors analysed the expression of *creA* (wild-type and modified sequences) by Northern blotting and small RNA-seq (Figure 2 and Suppl Fig 1). I wondered whether the *creT* expression could be monitored?

Response: Because *creT* transcription was repressed by *CreA*, we detected very few *creT* reads by RNA-seq, and could not tell whether they are innate or processed products (see below). Therefore, the RNA-seq data could not provide convincing information to determine the transcription features (like TSS and TTS) of *creT*. In this revision, we provided primer extension data to show the TSS of *creT* (Supplementary Fig. 3). Besides, we also performed Northern blotting to confirm the expression of *creT* (see the new Supplementary Fig. 1). Note that, there were several processed RNA products of different sizes, which need be further investigated.

Supplementary Fig. 1

3. Lines 130-149: before describing the repeat replacement, please explain and show how the toxicity of *creT* from this new TA module has been demonstrated as well as the antitoxin action of wt *creA*. *CreT* toxicity is shown only later in the manuscript (Fig 4 and Supplm Fig 8 for different

Acinetobacter isolates). To demonstrate the functional link with CRISPR-Cas machinery as for previously described creTA, the transformation efficiency in delta cas background is presented in Suppl Fig 2 and mentioned only at the end of the paragraph. I would suggest indicating these results from the beginning when describing the identification of a new CreTA associated with CRISPR-Cas.

Response: Thank you for this suggestion. In fact, CreT toxicity could be reflected from the transformation efficiency shown in Figure 2. To confirm the antitoxic function of CreA in a heterologous host, repeat replacement proved to be critical (*Nucleic Acids Research* **50**, 9442). The logic of current manuscript just accords with the actual exploring process.

As suggested by another reviewer, we include a new scheme showing the principle of the repeat replacement assay in Figure 2, to help readers understanding the functional link between CRISPR-Cas and CreTA and interpreting the data of repeat replacement assay.

4. An important remaining question is the nature of CreT. The authors tried to assess this point in the manuscript by mutational analysis but it remains still unclear if it acts as an RNA or as a small peptide translated from mini-ORF (“its protein product (if it does exist)” conclusion line 237-238)? I wondered whether the product/active translation could be detected in vitro (for example in vitro transcription/translation system) and in Acinetobacter (global approaches like riboprofiling or small peptides proteomics)?

Response: Thank you for this suggestion. Our data in Figure 4c (U4-G12 complementarity rather than their nucleotide identity was critical for toxicity) and Supplementary Fig. 5 (synonymous mutation subverted toxicity) clearly showed that CreT acts as an RNA rather than a protein (no matter this protein does exist or not). In fact, nearly all the characterized CreT RNA toxins from archaea conservatively have a mini-ORF, which may increase the turnover rate of CreT toxins (e.g., those sequestering the rare tRNA molecules), or perhaps encode short peptides of unknown function (but what is sure is that they are not related to toxicity according to our previous studies (*Science* **372**: eabe5601; *Nucleic Acids Res* **49**, 10677)).

As suggested, we performed small peptides proteomics using samples before or (10, 20, and 30 min) post CreT induction. Nevertheless, we did not detect the production of the predicted small peptide, but interestingly observed alterations in small peptides derived (digested) from ribosomal proteins (e.g., L30, L25, S21, etc.) and the tRNA-splicing ligase (data not shown), which support the disruption of protein synthesis and need be further interpreted combining other data (e.g., the transcriptomic data).

The authors checked the functionality of SD region associated with mini-ORF and discussed the results (lines 212-218) presented on Suppl Fig 3. On my opinion it is important to mention this earlier in the text when describing first gfp fusions (paragraph starting with line 164).

Response: The suggested paragraph discussed the transcriptional regulation of creT by CreA, while Supplementary Fig. 3 (Supplementary Fig. 4 in this revision) validated the function of creT SD motif. On our opinion, they are two irrelevant subjects. The suggested paragraph was revised for more clarity.

5. The use of gfp reporter gene fusion constructs should be better explained in the following paragraph “CreA transcriptionally repress its cognate creT”. It is not clear if transcriptional or

translational fusions are used in these experiments. To check the activity of promoter and transcriptional effect of *creA* on it, the authors should construct a transcriptional fusion carrying only promoter regulatory elements from the gene of interest and keeping canonical translational initiation signals for *gfp*. When using the constructs carrying both the promoter elements and SD to initiate translation both transcriptional and post-transcriptional/translational effects could be seen. The constructs presented should be combined with the data on Suppl Fig 3 with *ptac* promoter and mini-ORF SD.

Response: Thank you for this question. As suggested above, we provided new data (Supplementary Fig. 3) to determine the TSS of *creT*, which locates ~10 bp upstream of the target site of CreA (depicted in Figure 3a). That means Csy-CreA binding here may impede the initiation or the initial elongation of *creT* transcription. To faithfully assess this transcriptional effect, we have to include the “5’-UTR” sequence to include the target site of CreA.

The suggested paragraph “CreA transcriptionally repress its cognate *creT*” was substantially revised for more clarity. Notably, Csy-CreA in principle do not regulate the translation process. On one hand, type I CRISPR effector only recognize DNA target where the PAM motif (does not exist on RNA) plays a critical role for target recognition. On the other, the translational elements SD and start codon locate far away (~60 bp) from the target site of CreA. Related explanations were added in line 182-185.

As explained above, the data in the original Supplementary Fig. 3 (Supplementary Fig. 4 in this revision) was designed to characterize the SD motif of *creT*, which we think is irrelevant to its transcriptional regulation.

On my opinion, the paragraph starting from the line 174 should be modified. Fluorescence measurement and *gfp* RNA abundance analysis could not directly respond to the question about the translational effects. Please clarify this part including the Supplm Fig 3 constructs and additional combinations of regulatory elements.

Response: Revised as suggested. We agree that the question about the translational effects does not make sense, because type I CRISPR-Cas effectors can only recognize DNA targets (where the PAM sequence play critical roles in initiating target recognition). As explained above, Supplementary Fig. 3 (Supplementary Fig. 4 in this revision) is irrelevant to *creT* regulation and was not included in this part.

6. Line 183: the title should be modified: “does not have a canonical seed”

Response: Corrected correspondingly.

7. Lines 267-268: The authors concluded that the toxicity of CreT seems to correlate with the efficiency of translational initiation, but direct evidence for this correlation is still missing. On my opinion, it is important to try to detect the active translation or protein product to conclude or modulate the statement.

Response: This statement was modulated correspondingly. Our data in Figure 4c (U4-G12 complementarity rather than their nucleotide identity was critical for toxicity) and Supplementary Fig. 5 (synonymous mutation subverted toxicity) clearly showed CreT acts as an RNA rather than a protein, whether this protein does exist or not. In fact, nearly all the characterized CreT RNA

toxins from archaea conservatively have a mini-ORF, which may increase the turnover rate of CreT toxins (e.g., those sequestering the rare tRNA molecules), or perhaps encode a short peptide of unknown function (but not related to toxicity according to our previous studies (*Science* **372**: eabe5601; *Nucleic Acids Res* **49**, 10677)).

8. The paragraph “CreT is bactericidal small RNA likely targeting ribosomal RNAs” should be modified. On my opinion, the part on CreT toxicity (lines 281-287) should be moved to the first part of the manuscript for the description of TA module and the title of the paragraph should be modified accordingly.

Response: Thank you for this suggestion. It was the actual exploring process that we first identified one of the three candidates listed in Fig. 1 to be a real TA system (by assessing the toxicity based on transformation efficiency and the antitoxicity after repeat replacement, see the new Figure 2), and then dissected its essential functional elements (Figure 3 and 4). As suggested by another reviewer, we include a new scheme showing the principle of the repeat replacement assay in Figure 2, which helps readers to understand the functional link between CRISPR-Cas and CreTA and to interpret the data of the repeat replacement assay.

The authors claim that CreT interacts with 16S RNA and possibly with 23S rRNA (lines 278-279). I wondered whether the authors tried to identify the direct targets of CreT? To strengthen this conclusion the targets need to be identified experimentally. I would suggest at least including some elements on the CreT mechanism of action in the discussion section.

Response: We tried several methods to identify the targets and mechanisms of CreT. We first screened for colonies resistant to this toxin and subjected them to illumina sequencing, but unfortunately, we did not find any genetic variations on their genome (and unexpectedly, the resistant colonies became sensitive again after serial cultivation). We also performed transcriptomic analysis and found that nearly half of the tRNA genes (37 out of 72) were upregulated by > 2-fold after CreT induction, especially a gene of tRNA-Met (ABAYE_RS12755), which increased by 35-fold. However, over-expression of this tRNA gene could not relieve the toxicity (data not shown). These data supported that CreT does target the translation machinery; but the exact mechanism and target remain to be further investigated. The above information was included in Discussion part.

9. The authors claim that CreTA improved the efficiency of CRISPR antimicrobials (line 351-352), the difference is not very high (97.84% to 98.96%), is it statistically significant? Please specify.

Response: By calculating the killing efficiency for each replicate, we obtained the *P* value 2.61e-07 (specified in the new manuscript), which support the statistical significance of this difference. In fact, the log-transformed data differed by 0.4, which is a considerable increase for antimicrobial efficiency. In addition, from the plot in Figure 6c, statistical analysis showed that the *oxa23*-targeting CRISPR, when induced, resulted significantly fewer survivors in the presence of creTA.

10. For the application of this antimicrobial strategy it is important to assess the long-term effects through several generations during bacterial growth. Has the survival over the more long-term been tested?

Response: We agree it is important to assess the long-term effects through several generations during bacterial growth; however, at present it was difficult to make this assessment. Because

conjugation occurred much less efficiently in liquid cultures, and engineered *Acinetobacter baumannii* phages have not been currently developed. We are still attempting to engineer an *A. baumannii* phage to achieve this assessment.

Minor Issues:

- It could be helpful including page number to the manuscript.

Response: Added as suggested.

- The delay in providing supplementary data on submission was not helpful for the paper reviewing.

Response: We sincerely apologize for the delay.

- “x-log reduction” to avoid throughout the manuscript (Line 139, 144, 145, 192, 196, 200 ...)

Response: Thank you for this suggestion. We avoided this expression in some unnecessary places (see line 138 and 203 for examples).

- Line 60 “*Clostridium difficile*” is a previously used name, now “*Clostridioides difficile*” is currently used, please check

Response: Revised as suggested.

- Line 123 Please modify “Shine-Dalgarna” to “Shine-Dalgarno”

Response: Modified as suggested.

- Line 153 and 232 “wild-type”

Response: Revised as suggested.

- Line 296 “(bearing an *aac3*-targeting CRISPR)” please specify “mini-CRISPR arrays with spacer targeting the *aac3*”/specific crRNA ? The same for line 338 “*oxa23*-targeting CRISPR”.

Response: Revised as suggested.

- Line 298 Please explain why “potassium tellurite” is used for selecting for all transformants.

Response: To deliver the CRISPR antimicrobial into target pathogen, we employed a conjugative plasmid carrying the potassium tellurite-resistant gene as a selection marker. This information has been provided in figure legends, main text and in methods part.

- Fig. 1 Please include the position of promoter region/TSS/terminator for *creA* and *creT*.

Response: Included as suggested (Figure 1). As stated above, we could not get convincing evidence for the terminator from current data, and did not include this information (which is dispensable for this study).

- Line 621 “10 mg RNA” please check, I think it should be “10 microgram”.

Response: Revised as suggested.

Response to Reviewer #3:

In the manuscript entitled "ATTACK: Associate Toxin-Antitoxin with CRISPR-Cas to Kill Multidrug-Resistant Pathogens", Wang et al reported and characterized the first bacterial CreTA system, established its working mechanism, and applied the system to boost the CRISPR-mediated antimicrobials in *A. baumannii*. The work is highly innovative with both new fundamental principles and substantial translational values. I have a few points for the authors.

Response: Much appreciation for your positive evaluations and helpful suggestions.

1. The authors only characterised the system and applied the system in *A. baumannii*, rather than several common MDR pathogens, and it is not straightforward to extend the system to other pathogens. Hence, it would be appropriate to change the title to "...kill Multidrug-Resistant *Acinetobacter Baumannii*".

Response: Thank you for this suggestion. Because CRISPR antimicrobials have proved to be effective in killing various MDR pathogens, and our recent studies showed that CreTA widely distributes in bacteria and archaea (*Science* **372**: eabe5601; *Nucleic Acids Res* **49**, 10677), the combined strategy of CRISPR and CreTA we demoe here using the priority AR pathogen CRAB, should be applicable to treat other pathogens.

2. A large number of CRISPR-Cas systems in bacteria are degenerated or are inhibited by Acrs. My understanding is that the ATTACK system works only in those pathogens containing an active CRISPR-Cas system, not those with a degenerated CRISPR-Cas system or those containing Acrs. Authors should clarify this point.

Response: CRISPR antimicrobials usually directly deliver the CRISPR-Cas encoding genes into the target pathogens, which do not necessarily have an endogenous active CRISPR-Cas system. As to Acr proteins that may inactivate the CRISPR antimicrobials, our ATTACK strategy was just designed to sense and combat these proteins: upon CRISPR inactivation, CreTA will be triggered to kill the pathogens.

3. The authors claim that ATTACK is applied to eliminate MDR pathogens, what are the MDR profiles of the strains they used in this study.

Response: The MDR profiles of these strains were provided in the new Supplementary Data 1.

4. What is the potential of the application of ATTACK system in vivo? Authors may discuss this point in the Discussion.

Response: Discussed as suggested (line 422).

5. A schematic illustration of the working principle of the test should be provided in Fig 2.

Response: Provided as suggested (see below).

Figure 2

Minor points:

There are grammar and sentence construction problems in the text, I am listing some of them:

1) L 30 and L354-355, "kill pathogens and cure resistance genes" can not be written in short as "kill or cure MDR pathogens".

Response: Revised correspondingly.

2)L 44, appearance should be "emergence"

Response: Corrected.

3)L 58, cure them, "them" mean pathogens or infection? Infections can be cured, but not pathogens.

Response: Corrected.

4) L124, runs should be "is oriented"

Response: Corrected.

5) L135, make should be "enable"

Response: Corrected.

6) L227, decide should be "dictate"

Response: Corrected.

7) 253, disappeared should be "was lost"

Response: Corrected.

8)L 277, kept should be "retained"

Response: Corrected.

9)L312, partly should be "partially"

Response: Corrected.

10) L338, put should be "place"

Response: Corrected.

Reviewers' Comments:

Reviewer #1:

Remarks to the Author:

The authors satisfactorily addressed the concerns I raised.

Reviewer #2:

Remarks to the Author:

Thank you for the revision of the paper and additional experiment included for creT detection/TSS identification with corresponding supplementary figure. Most of my concerns has been addressed in the revised version of the manuscript.

Reviewer #3:

Remarks to the Author:

The authors adequately addressed my points.

Point-by-point Response to Referees

Response to Reviewer #1:

The authors satisfactorily addressed the concerns I raised.

Response: We sincerely appreciate all your kind comments and constructive suggestions.

Response to Reviewer #2:

Thank you for the revision of the paper and additional experiment included for creT detection/TSS identification with corresponding supplementary figure. Most of my concerns has been addressed in the revised version of the manuscript.

Response: We sincerely appreciate all your kind comments and constructive suggestions.

Response to Reviewer #3:

The authors adequately addressed my points.

Response: We sincerely appreciate all your kind comments and suggestions.